# Malignant extracellular vesicles carrying *MMP1* mRNA facilitate peritoneal dissemination in ovarian cancer

Akira Yokoi[1,2], Yusuke Yoshioka[1], Yusuke Yamamoto[1], Mitsuya Ishikawa[3], Shun-ichi Ikeda[3], Tomoyasu Kato[3], Tohru Kiyono[4], Fumitaka Takeshita[5], Hiroaki Kajiyama[2], Fumitaka Kikkawa[2] & Takahiro Ochiya[1]

Advanced ovarian cancers are highly metastatic due to frequent peritoneal dissemination, resulting in dismal prognosis. Here we report the functions of cancer-derived extracellular vesicles (EVs), which are emerging as important mediators of tumour metastasis. The EVs from highly metastatic cells strongly induce metastatic behaviour in moderately metastatic tumours. Notably, the cancer EVs efficiently induce apoptotic cell death in human mesothelial cells *in vitro* and *in vivo*, thus resulting in the destruction of the peritoneal mesothelium barrier. Whole transcriptome analysis shows that *MMP1* is significantly elevated in mesothelial cells treated with highly metastatic cancer EVs and intact *MMP1* mRNAs are selectively packaged in the EVs. Importantly, *MMP1* expression in ovarian cancer is tightly correlated with a poor prognosis. Moreover, *MMP1* mRNA-carrying EVs exist in the ascites of cancer patients and these EVs also induce apoptosis in mesothelial cells. Our findings elucidate a previously unknown mechanism of peritoneal dissemination via EVs.

[1] Division of Molecular and Cellular Medicine, National Cancer Center Research Institute, 5-1-1 Tsukiji, Chuo-ku, Tokyo 104-0045, Japan. [2] Department of Obstetrics and Gynecology, Nagoya University Graduate School of Medicine, 65 Tsuruma-cho, Showa-ku, Nagoya 466-8550, Japan. [3] Department of Gynecology, National Cancer Center Hospital, 5-1-1 Tsukiji, Chuo-ku, Tokyo 104-0045, Japan. [4] Division of Carcinogenesis and Cancer Prevention, National Cancer Center Research Institute, 5-1-1 Tsukiji, Chuo-ku, Tokyo 104-0045, Japan. [5] Department of Functional Analysis, Fundamental Innovative Oncology Core Center (FIOC), National Cancer Center Research Institute, 5-1-1 Tsukiji, Chuo-ku, Tokyo 104-0045, Japan. Correspondence and requests for materials should be addressed to T.O. (email: tochiya@ncc.go.jp).

Ovarian cancer is the most lethal reproductive system cancer and a leading cause of cancer-related death[1]. In 2010, 160,500 patients died from this cancer worldwide and this number is rapidly increasing[2,3]. The poor prognosis and survival outcomes of patients have not been significantly altered in recent decades. More than 75% of ovarian cancer patients are diagnosed at an advanced stage because of the lack of both specific clinical symptoms and effective early detection screening. In addition, the 5-year survival rate of these patients is ∼20% (ref. 4). Metastasis to the abdominal cavity is frequently observed in ovarian cancer patients and is one reason for the unfavourable outcomes and poor prognosis[5]. Ovarian cancer is disseminated at a very early phase and it is extremely difficult to overcome and control this metastasis[6]. Despite ongoing basic research, the detailed mechanism of peritoneal dissemination in ovarian cancer remains unknown. Thus, it is critical to understand the underlying molecular mechanisms, which may ultimately improve patient outcomes.

Recent evidence has demonstrated that cancer cells secrete extracellular vesicles (EVs) to both proximal surrounding cells and distal sites, thereby enabling the development of a cancer microenvironment that in turn promotes cancer invasion and metastasis[7–12]. In general, EVs, including exosomes and microvesicles, are small membrane vesicles that contain various bioactive molecules, such as microRNAs (miRNAs), messenger RNAs and proteins[13–17]; they are released from all cell types and play key physiological roles in intercellular communication[18–20]. Ovarian cancer cells aggressively migrate into the peritoneal cavity and the ascetic fluid provides a favourable environment for wide dissemination[21]. Given the pathophysiological functions of EVs in cancer cells and their microenvironment, and the fact that EVs fully demonstrate those abilities in the presence of humoral factors[22,23], it is highly plausible that ovarian cancer-derived EVs in ascites contribute to tumour progression and subsequent peritoneal dissemination.

Here we demonstrate that EVs derived from highly metastatic ovarian cancer cells promote peritoneal dissemination in vivo. In the present study, to acquire novel insights into the peritoneal metastasis of ovarian cancer cells, we investigated how the EVs derived from cancer cells affect mesothelial cells, which are one of the main barriers between cancer cells and the peritoneal cavity. We also identify a key molecule in the EVs that promotes tumour progression. These results provide a premise for the understanding of a previously unknown mechanism for peritoneal dissemination that is associated with cancer-derived EVs.

## Results

**Establishment of mouse models for peritoneal dissemination.** Ovarian cancer cell lines are usually established from metastatic tumours, but an intercellular comparison of these metastatic phenotypes of these cells has not been conducted to date. To identify the molecular mechanisms of peritoneal dissemination in ovarian cancer, we established a mouse model for this type of metastasis. A small incision was made on the left back of a mouse and the left ovaries were pulled out. Then, four types of ovarian cancer cells (serous types: SKOV3 and A2780; clear cell types: RMG-1 and ES-2) were orthotopically transplanted into the left ovaries (Fig. 1a). All cell lines stably expressed luciferase; thus, the tumour progression was non-invasively monitored by an in vivo imaging system (IVIS) every week. Using the IVIS, we observed increased bioluminescence in all cell lines. When the mice were dissected, we found that the primary left ovarian tumours were enlarged in all animals, and that metastatic tumours were found in the peritoneal cavities in ES-2-, SKOV3- and A2780-transplanted mice but not in RMG-1-transplanted mice (Fig. 1b–d).

Interestingly, analysis of the tumour progression indicated that the aggressiveness was different among the four ovarian cancer cell lines (Fig. 1e). For example, ES-2 cells resulted in fatal peritoneal dissemination in only 2 weeks, whereas RMG-1 cells did not produce any metastatic tumours. A2780 cells and SKOV3 cells were also metastatic, but they required a longer time for peritoneal dissemination than ES-2 cells did. Thus, these models recapitulated early-stage ovarian cancer progression at different rates.

**EVs from high-metastatic cells promote metastasis in vivo.** As emerging and accumulating evidence has suggested that tumour-derived EVs are critical mediators of tumour progression and metastasis, we hypothesized that cancer-derived EVs could influence the status of peritoneal dissemination. Our data indicated that the EVs from ES-2 cells, a highly and rapidly metastatic cell line, possessed the highest malignant potential. Therefore, we next examined whether malignant EVs accelerated peritoneal dissemination in vivo. To this end, luciferase-expressing A2780 cells, which displayed a moderate metastatic ability (Fig. 1d,e), were orthotopically injected into the left ovaries and 5 μg of EVs from ES-2 (highly metastatic) cells, RMG-1 (non-metastatic) cells and immortalized human ovarian surface epithelium 1 (HOSE1) cell lines (as a control)[24] and phosphate-buffered saline (PBS) were intraperitoneally injected six times (Fig. 2a). The EVs used in these experiments were isolated with ultracentrifugation and characterized by phase-contrast electron microscopy and a nanoparticle tracking system (Supplementary Fig. 1a,b). Further basic characterization of the EVs was performed (Supplementary Fig. 1c,d). To evaluate the amount of metastatic tumours, we measured the bioluminescent signal intensities in the peritoneal cavity using IVIS after removing the primary tumours (Fig. 2b). In contrast to the primary tumour size, which was not changed by the treatments (Fig. 2c and Supplementary Fig. 2a), the mice injected with ES-2 EVs had a significantly greater metastatic burden in the peritoneal cavity compared with the mice injected with HOSE1 EVs or with RMG-1 EVs. To further examine the ability of EVs involved in peritoneal metastasis, the same experiments using the mice transplanted with either SKOV3 cells or RMG-1 cells were performed (Supplementary Fig. 2b,c). As a result, the metastasis in the mice with SKOV3 was significantly promoted by ES-2 EVs, but metastasis in RMG-1 was not changed after successive EV injections. These data suggested that the high metastatic EVs could only promote metastasis but do not change the property of cancer cells from non-metastatic to metastatic.

To further investigate the effects of EVs on peritoneal dissemination, we established neutral sphingomyelinase 2 (nSMase2) knockdown (KD) cell lines using A2780 cells and ES-2 cells. nSMase2 is an EV secretion-related protein and KD of nSMase2 caused a significant decrease in EV secretion[12]. Orthotopic mouse models were generated with ES-2- and A2780-nSMase2 KD cells (Supplementary Fig. 3a), and metastasis was assessed by IVIS. As expected, mice with both nSMase2 KD cell lines had a significant reduction in metastasis in the peritoneal cavity compared with nSMase2-negative control cells (for ES-2 and A2780 cells, mice were killed 2 and 4 weeks after injection, respectively), but there was no difference in the primary tumour size (Fig. 2d and Supplementary Fig. 3b).

To determine the effect of ES-2 EVs on peritoneal dissemination in detail, we tested the dose dependency of the EV injection (two, four and six times) in the same experimental model (Fig. 2e and Supplementary Fig. 3c). Although a subtle increase in metastasis was observed after four injections, six injections obviously promoted metastasis, suggesting that the amount and

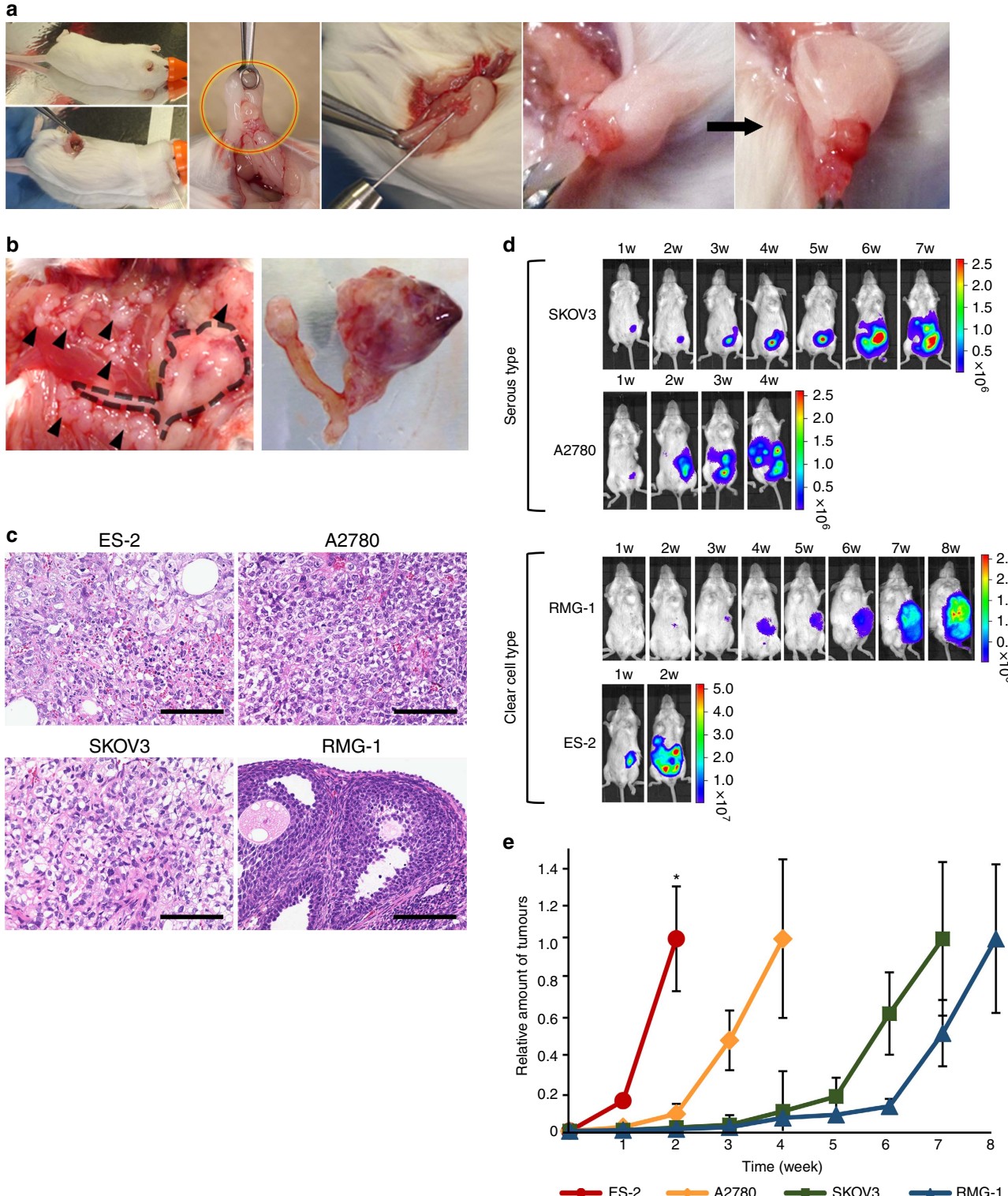

**Figure 1 | Establishment of mouse models for peritoneal dissemination in ovarian cancer.** (**a**) Illustrative photos of an orthotopic mouse model of ovarian cancer. A small incision (around 1 cm) was made on the left back of the mice. The ovaries were pulled out (red–yellow circle) and $1 \times 10^6$ cells resuspended in 50 μl of PBS were injected into left ovarian bursa. (**b**) Representative photographs at the time of killing. The left photo shows metastatic tumours (black arrowheads) and the uterus with the primary left ovarian tumour (a black dotted line indicated the position), covered with fat pad. The right photo represents the dissected uterus with the primary left ovarian tumour. (**c**) Histological features of primary tumours in mouse model. The tumours were obtained from the orthotopic mouse model by using ES-2, A2780, SKOV3 and RMG-1 cells. Haematoxylin and eosin (H&E) staining. Scale bars, 100 μm. (**d**) Representative bioluminescence images of orthotopic mouse models. The mouse models were established by four types of ovarian cancer cell lines. Tumour progression was monitored weekly by an IVIS. (**e**) Quantitative analysis of tumour progression. To describe the differences in progression, the photon values were compared with the value at the time of killing, which was set to 1.0. $n = 3$. Error bars represent s.d. There were significant differences at the 2 weeks time point by using analysis of variance followed by the Tukey–Kramer test for ES-2 versus other (*$P < 0.05$).

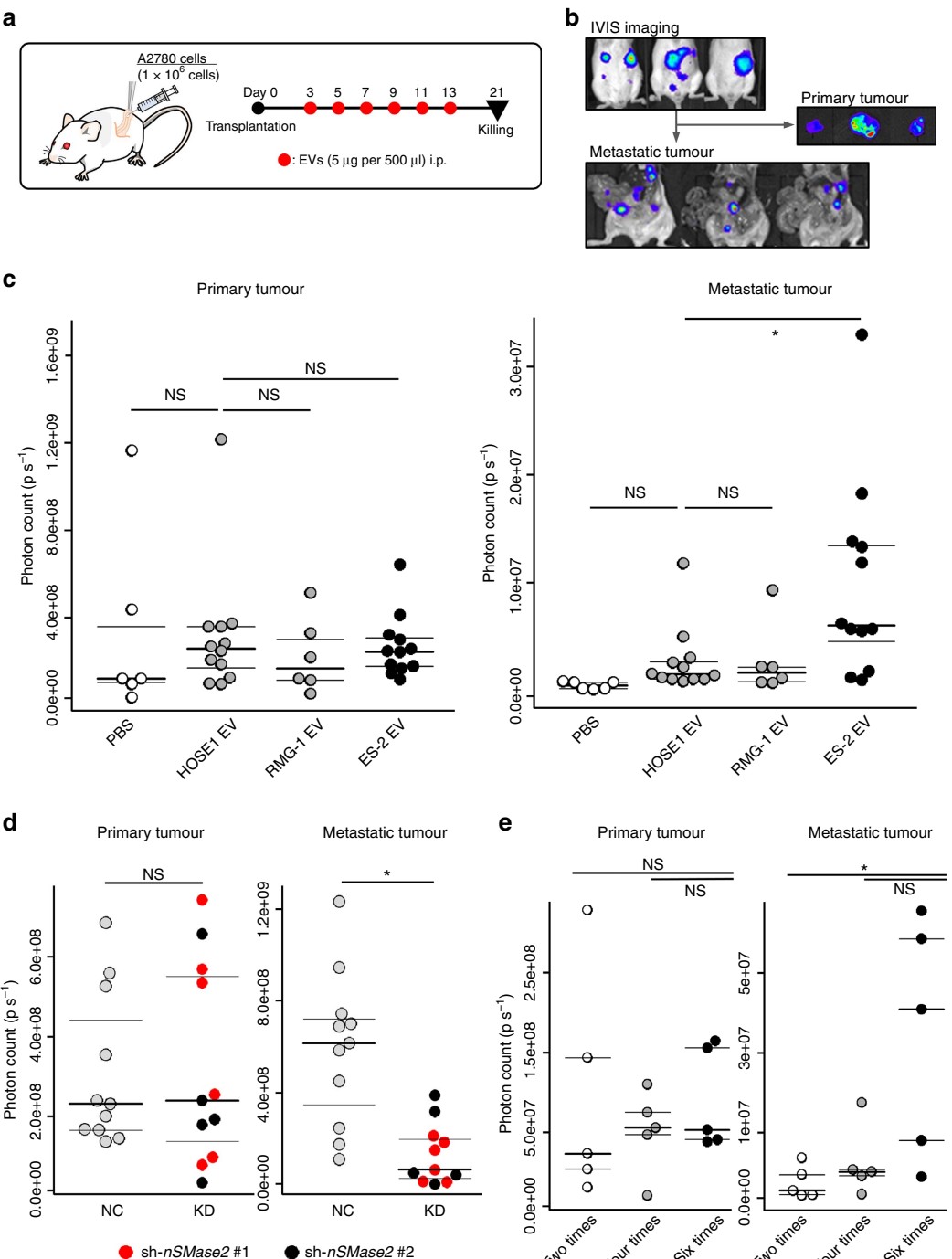

**Figure 2 | EVs derived from highly metastatic ovarian cancer cells promote peritoneal dissemination *in vivo*.** (**a**) Schematic protocol for investigating the EVs in peritoneal dissemination. Orthotopic mouse models were established with A2780 cells. ES-2 EVs, RMG-1 EVs, HOSE1 EVs and PBS were injected i.p. from day 3 to 13, every other day, 6 times. The mice were killed on day 21. This protocol was applied in **c**. (**b**) Representative bioluminescence images of orthotopic mouse models at day 21. The upper images were captured from outside the bodies using the IVIS. The right-middle images show dissected primary tumours. The lower images indicate the metastatic tumours in the peritoneal cavity after the primary tumours in these mice were already removed. (**c**) Distribution of photon count in the dissected primary tumour (left) and the peritoneal metastatic tumours (right). The mice were treated with three types of EVs and PBS as described in **a**. $n = 12$; ES-2 EVs, 6; RMG-1 EVs, 12; HOSE1 EVs and 6; PBS. Dunnett's test (*$P < 0.05$ and NS, no significance). (**d**) Distribution of photon count in the dissected primary tumours (left) and the peritoneal metastatic tumours (right) using *nSMase2* KD cells and negative control (NC) cells. The orthotopic mouse model was established by injection with ES-2 cells and the cells were transfected with the two different sequences of *nSMase2* shRNA vector (KD; 1 and 2) or control vector (NC). The mice were killed at day 14. KD, knockdown of *nSMase2*; NC, negative control. Student's *t*-test (*$P < 0.05$ and NS, no significance). (**e**) Dose dependency of ES-2 EVs on promoting peritoneal dissemination. Orthotopic mouse models were established with A2780 cells. ES-2 EVs were injected i.p. from day 3 and every other day thereafter. The names of the groups indicated how many times the EVs were injected. The mice were killed on day 21. Distribution of photon count in the dissected primary tumour (left) and the peritoneal metastatic tumours (right). $n = 5$; 2 times, $n = 5$; 4 times, $n = 5$; 6 times. Dunnett's test (*$P < 0.05$ and NS, no significance).

timing of the EV injection are critical for the induction of peritoneal metastasis, at least in our model. Collectively, these data clearly demonstrate that the EVs derived from metastatic ovarian cancer cells, in particular ES-2 cells, promote peritoneal dissemination *in vivo*.

**Induction of apoptosis in mesothelial cells by EVs *in vivo*.** Although EVs are known to be involved in communication between cancer cells and the surrounding microenvironment such as endothelial cells and fibroblasts, the peritoneum meso-thelial cell layer is the most important anatomical barrier for preventing the peritoneal dissemination of ovarian cancer cells. Here we hypothesized that EVs derived from high metastatic cancer cells could have some effects on mesothelial cells, because the peritoneum is the major obstacle that cancer cells must overcome for metastatic progression (Supplementary Fig. 4). Therefore, the effect could be favourable for metastatic cancer cells in terms of demolishing the barrier, when the cells drop into the ascites and when it creates metastatic sites. Thus, we evaluated those effects from cancer EVs to mesothelial cells. Peritoneal mesothelial cells were stained for mesothelin, a specific marker for the mesothelium, and a single layer of cells covering the peritoneal cavity was observed (Fig. 3a). PKH67-labelled EVs were intraperitoneally injected into severe combined immuno-deficiency (SCID) mice (Fig. 3b) and the incorporation of the EVs was stereomicroscopically observed in the mesothelial layer of the abdominal wall, liver surface and omentum 12 h after the EV injection (Fig. 3c and Supplementary Fig. 5). To further confirm the labelled EV uptake and exclude the possibility that labelled EVs simply attached to the surface, the mice were dissected and frozen sections were prepared from the abdominal wall. PKH67 fluorescence was detected in the mesothelial cells (Fig. 3d). These findings indicate that EVs are incorporated into the peritoneal mesothelial cells *in vivo*.

To further analyse the effects of EVs *in vivo*, ES-2 EVs were intraperitoneally injected into SCID mice (each 30 µg, every 12 h, 5 times) and the morphology of the peritoneal wall was assessed after the mice were killed 6 h following the last injection. In contrast to treatment with the PBS control, the smooth structure of the peritoneum membrane and the surface of the organs were partly collapsed in the mice treated with ES-2 EVs (Fig. 3e). Moreover, terminal deoxynucleotidyl transferase-mediated dUTP nick-end labelling (TUNEL)-positive cells were detected in the area damaged by treatment with the ES-2 EVs (Fig. 3e), indicating that ES-2 EVs induced apoptosis in mesothelial cells. A scanning electron microscopy (SEM) analysis was performed to directly observe the surface of the peritoneum. Surprisingly, we found many damaged sites that did not have microvilli (Supplementary Fig. 6a, red circles); submesotheilal connective tissue was visible in the peritoneal membrane treated with the ES-2 EVs (Fig. 3f and Supplementary Fig. 6a,b). An image showing the detachment of the cells indicated that the mesothelial cells were exfoliated (Supplementary Fig. 6c). Quantitative analysis showed that the number of defective sites was significantly greater than those of the two controls (Fig. 3g). These results supported our hypothesis that ES-2 EVs damaged the peritoneal mesothelium and promoted subsequent metastasis.

**Time-lapse observation of mesothelial cells damaged by EVs.** The *in vivo* observations of mouse mesothelium severely damaged by ES-2 EV exposure prompted us to conduct time-lapse imaging of human mesothelial cells with ES-2 EVs, to further elucidate the effects on human peritoneal dissemination. As a first step, we confirmed the uptake of EVs into two types of human mesothelial cells, MeT-5A cells and human peritoneal mesothelial cells

(HPMCs), using PKH67-labelled EVs and the fluorescence was observed by confocal microscopy (Fig. 4a,b and Supplementary Fig. 7). Next, to investigate the effects of ES-2 EVs on human mesothelial cells *in vitro*, we applied 30 µg per well of ES-2 EVs to MeT-5A cells and observed the cell morphology (Fig. 4c). Although the MeT-5A cells originally had a cobblestone shape, they gradually became spindle-shaped 48 h after the addition of ES-2 EVs, whereas the HOSE1 EVs did not induce morphological changes (Fig. 4d, left two images and Supplementary Movie 1). The number of cells with abnormal shapes significantly increased (Fig. 4d, right graph). We then conducted the same experiment using HPMCs (primary cultures of HPMCs), because confluent cultured HPMCs are typically used to mimic the peritoneum[25,26]. In this experiment, HPMCs were cultured to confluence in dishes and followed by the addition of either ES-2 or HOSE1 EVs (Fig. 4e and Supplementary Movie 2). Consistent with the MeT-5A experiments, ES-2 EVs caused morphological changes in HPMCs; gaps between the cells, called open areas, appeared (Fig. 4e, purple area). Quantitative analysis revealed that significantly greater gaps were observed following treatment (Fig. 4e, a right graph) and these gaps resembled the interstitial space, which could be favourable for cancer invasion. Collectively, these findings indicated that the highly metastatic ES-2 EVs severely damaged human mesothelial cells, suggesting that the EVs facilitate the peritoneal dissemination of ovarian cancer cells. These results were consistent with those of the *in vivo* experiments.

**MMP1 mRNA is a key molecule for the destructive phenotypes.** The cargo of EVs, which includes miRNAs, mRNAs, proteins and other biological components, is transferred from the donor to recipient cells and influences the cellular phenotypes. Therefore, we next investigated which genes were altered in the mesothelial cells treated with the metastatic EVs. The EVs from three meta-static cell lines ES-2, A2780 and SKOV3, as well as the controls, HOSE1 cells and HOSE2 cells, were added to MeT-5A cells. Total RNA was extracted from EV-treated MeT-5A cells, and micro-array analyses were performed (Fig. 5a). As shown in a heat map displaying the differentially expressed genes among treat-ments with cancer EVs and control EVs, there were obvious differences in gene expression in the MeT-5A cells (Fig. 5b and Supplementary Fig. 8a). Gene set enrichment analysis (GSEA) was performed for the ES-2 EV and HOSE1 EV treatments (Fig. 5c), to examine the effect of ES-2 EVs on mesothelial cells. Several pathways such as proteolysis and apoptosis were sig-nificantly enriched in ES-2 EV-treated MeT-5A cells. Similarly, GSEA was used to assess the A2780 versus HOSE1 and SKOV3 versus HOSE1 treatments. However, proteolysis and apoptosis enrichment was not found in these comparisons (Supplementary Table 1). To determine the key molecules involved in the phe-notypes of mesothelial cells, highly upregulated genes (top 12 genes) in ES-2 EV-treated MeT-5A cells were selected (Fig. 5d) and the expression changes in the microarrays were validated by quantitative reverse transcriptase–PCR (qRT–PCR) using inde-pendent sample sets (Supplementary Fig. 9a). Among the 12 genes, *MMP1* mRNA was validated at different doses and time points of the EV treatment (Supplementary Fig. 9b), and the upregulation of *MMP1* was validated in ES-2 EV-treated HPMCs (Supplementary Fig. 9c). *MMP1* was also included in the pro-teolysis pathway identified by GSEA (Fig. 5c). To further inves-tigate the relevance of *MMP1* in mesothelial cells treated with various EVs, an additional analysis was performed using the data from a microarray analysis (Supplementary Fig. 8b–e). When we looked at gene expression change in mesothelial cells treated with low or high metastatic EVs, *MMP1* was also contained in the

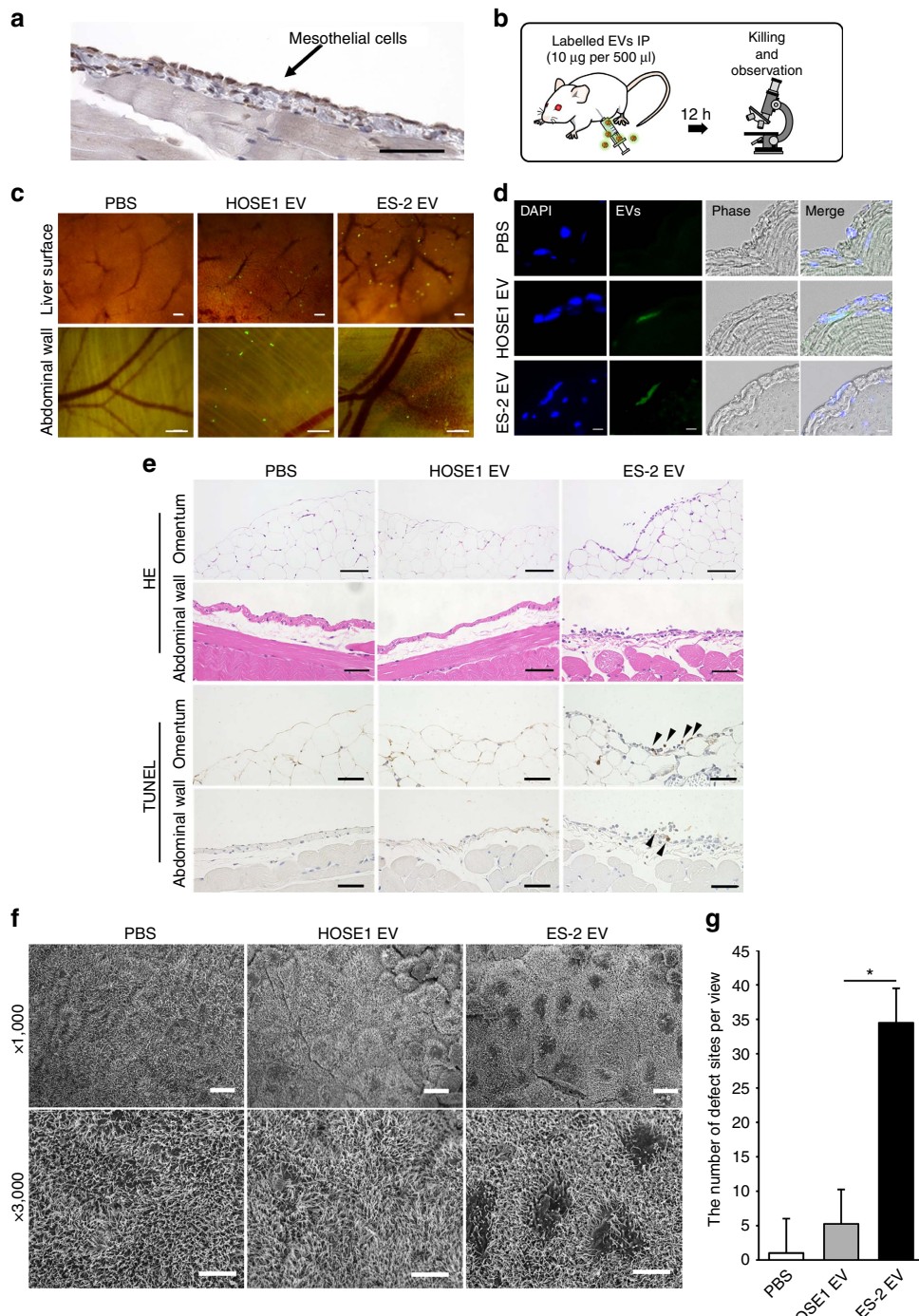

**Figure 3 | The EVs from highly metastatic cell lines damage the peritoneum and induce apoptosis *in vivo*.** (**a**) Representative slide of a mouse peritoneal membrane (single-layered mesothelial cells) that was immunostained with mesothelin. The abdominal wall was dissected from the mouse. Scale bar, 100 μm. (**b**) Schematic protocol for the uptake of EVs into the peritoneal membrane. The EVs from ES-2 cells were labelled with PKH67 and 20 μg was injected i.p. (**c**) Stereomicroscopic images showing labelled EVs that were incorporated into the peritoneum. Mice were killed after i.p. injection of PKH67-labelled EVs (20 μg). Green dots indicate EVs and the peritoneum on the liver surface and abdominal wall was observed. For the control, the same amount of PBS labelled by PKH67 green was used. Scale bars, 500 μm. (**d**) Representative microscopic section images of the abdominal wall treated with labelled EVs (DAPI: nuclei, green: EVs). The sections were cut to a thickness of 10 μm. Scale bars, 10 μm. (**e**) Representative microscopic images of HE-stained and TUNEL-stained slides from the abdominal wall and omentum. EVs were injected i.p. into mice (30 μg of EVs with 500 μl of PBS/12 h, 5 times). Black arrowheads indicate positive cells. Scale bars, 100 μm. (**f**) Representative SEM images of the abdominal wall treated with EVs. The EVs were injected i.p. into mice (30 μg of EVs with 500 μl of PBS/12 h, 5 times). The surface of the peritoneum is covered with numerous microvilli. Scale bars, 10 μm (upper 3 images) and 5 μm (lower three images). (**g**) Quantitative analysis of the damaged sites from SEM images. The defective sites were counted from 1,000 images that were randomly selected. Error bars represent s.d. Student's *t*-test (*$P < 0.01$). The data are representative of at least three independent experiments. **c**,**d**,**f**,**g**).

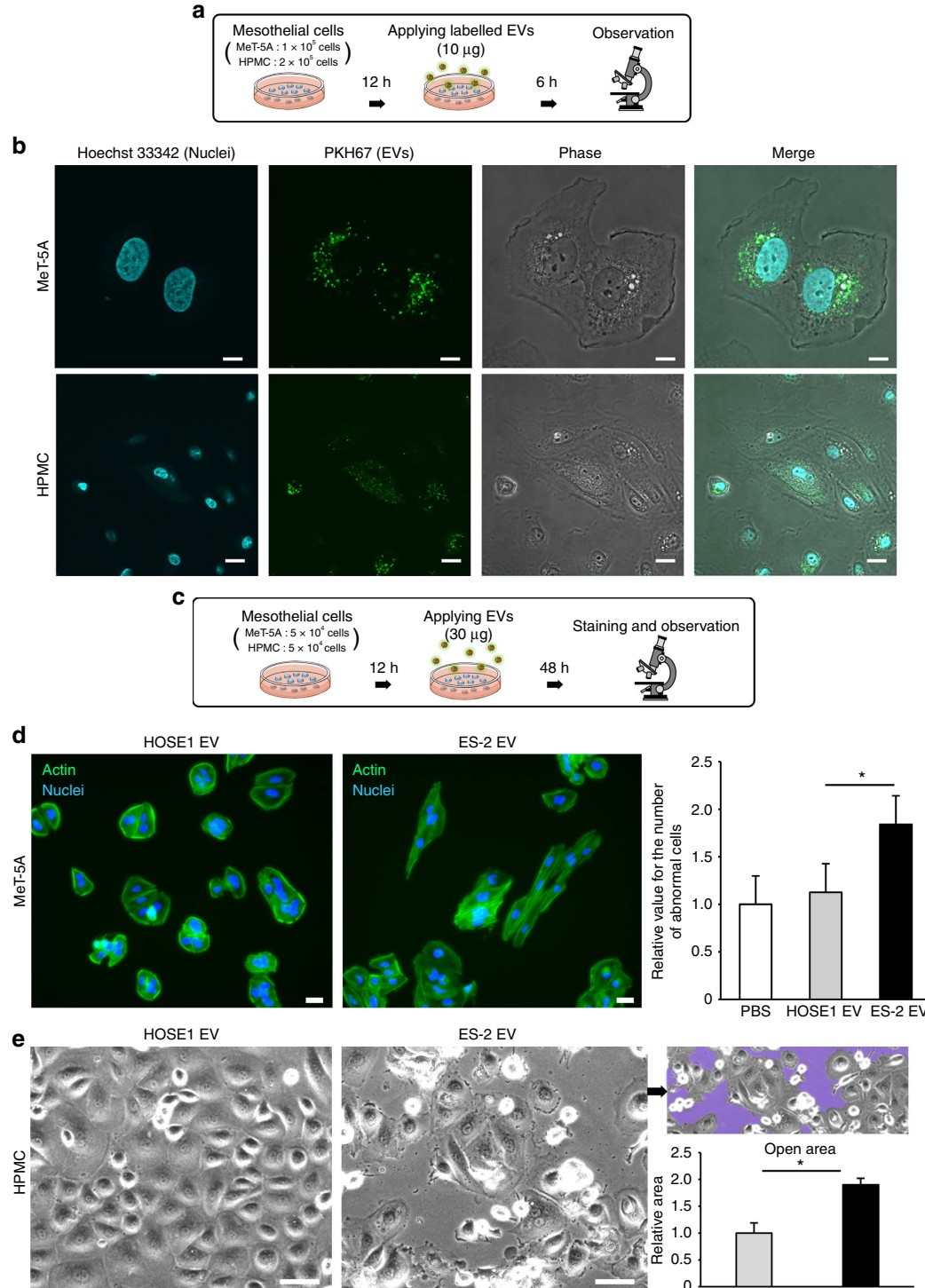

**Figure 4 | The EVs from highly metastatic ovarian cancer cell lines damage human mesothelial cells *in vitro*.** (**a**) Schematic representation of the uptake of EVs into the mesothelial cells. PKH67 green-labelled EVs were added to mesothelial cells and observed by confocal microscopy after 6 h. (**b**) Representative confocal microscopic images. EVs derived from ES-2 cells were labelled with PKH67 green and added to two types of human mesothelial cells (MeT-5A and HPMC). Scale bars, 10 μm. (**c**) Schematic representation of the protocol for analysing the effects of EVs on mesothelial cells. (**d**) Analysis of the effects of EVs on human MeT-5A mesothelial cells. EVs were added to MeT-5A cells and observed after fluorescence staining of the actin filaments (green) and nuclei (blue). The number of abnormally shaped cells was measured and is shown in the bar chart. The number of cells treated with PBS was set to 1.0 and other values were normalized to this number. Scale bars, 20 μm. Error bars represent s.d., Student's *t*-test (*$P < 0.01$). (**e**) Analysis of the effect of EVs on HPMCs. EVs were added to HPMCs and observed after 48 h. The open areas are shown in purple and the area was measured and is shown in the bar chart. Scale bars, 50 μm. Error bars represent s.d., Student's *t*-test (*$P < 0.01$). The data are representative of at least three independent experiments (**d,e**).

population of upregulated genes with high metastatic EVs (Supplementary Fig. 8b,c). Furthermore, *MMP1* is one of the specifically upregulated genes in mesothelial cells treated with highly metastatic EVs, ES-2 EVs (Supplementary Fig. 8d,e).

It is unclear whether the *MMP1* mRNA in the EVs are directly transferred to MeT-5A cells or whether *MMP1* is newly expressed in MeT-5A by the EV stimulus. For this purpose, we conducted RT–PCR of ES-2 EVs and found that the ES-2 EVs contained

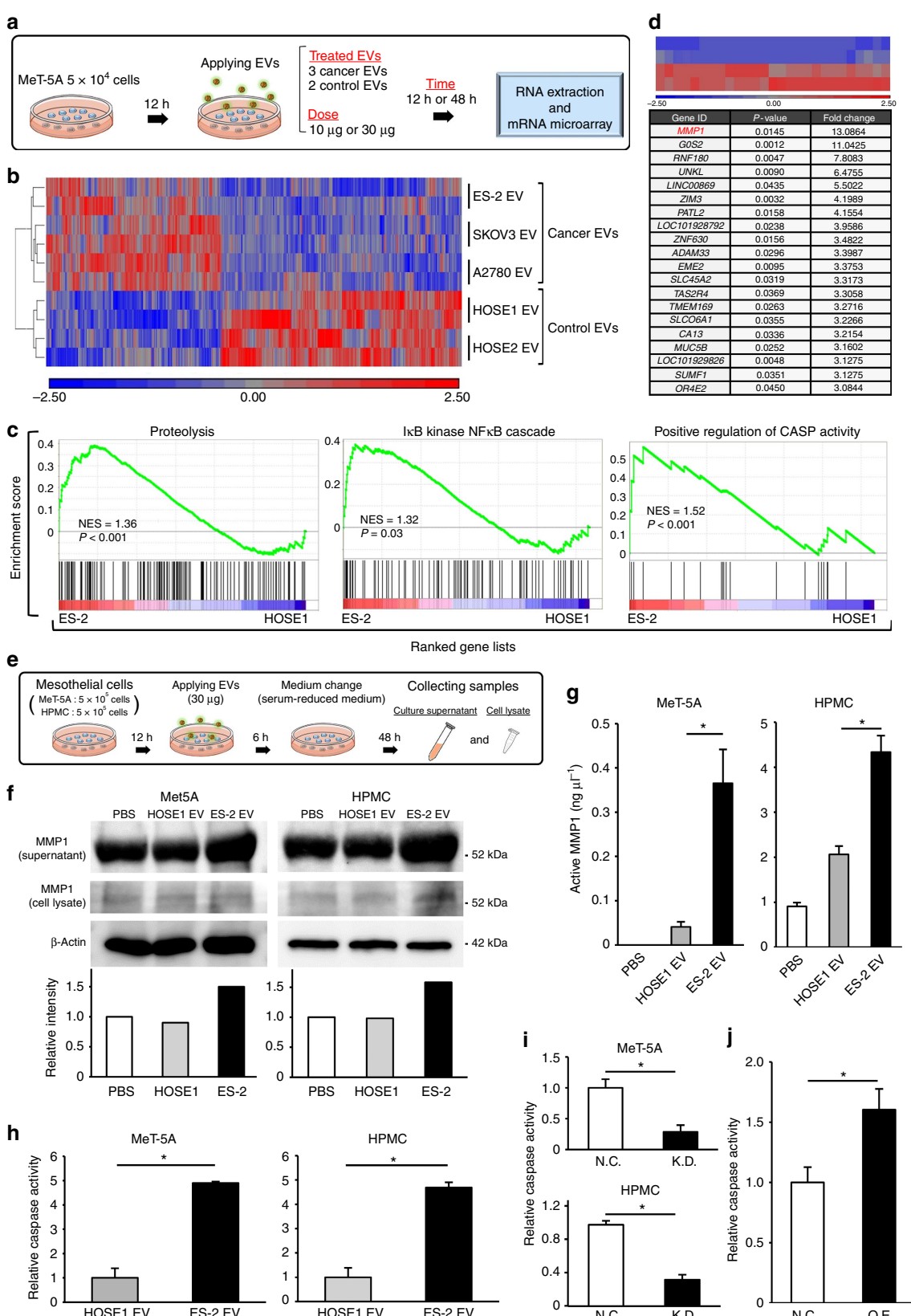

more *MMP1* mRNAs compared with other EVs (Supplementary Fig. 10b). Furthermore, by designing PCR primers covering the full-length *MMP1* gene, we found that intact *MMP1* mRNA was present in the ES-2 EVs (Supplementary Fig. 10c). According to our qRT–PCR quantification, although the amount of *MMP1* mRNAs in ES-2 cells was approximately 25 times greater than that of other cells, the *MMP1* amount in ES-2 EVs was more than 70 times enriched compared with the others (Supplementary Fig. 10a,b). This difference indicated that *MMP1* mRNAs were concentrated in the EVs, rather than in the cells. To determine whether the *MMP1* mRNA in EVs was functional, we next determined the MMP1 protein levels in mesothelial cells. ES-2 EVs and HOSE1 EVs were added to two types of mesothelial cells and their culture supernatant was changed to serum-reduced medium 48 h before collecting samples (Fig. 5e). Immunoblot analysis revealed that the addition of ES-2 EVs increased the amount of MMP1 protein in the culture supernatant of the mesothelial cells (Fig. 5f). enzyme-linked immunosorbent assay assays of MMP1 activity were performed and the concentration of active MMP1 protein in the cell culture supernatant was found to be significantly elevated by treatment with ES-2 EVs (Fig. 5g).

According to the results of the pathway analysis (Fig. 5c), the apoptosis pathway was significantly enriched in mesothelial cells treated with ES-2 EVs; thus, we measured the caspase activity in the two types of human mesothelial cells. Consistent with the *in vivo* observation of caspase activity (Fig. 3f), the treatment with ES-2 EVs yielded a significant increase in caspase activity in both types of human mesothelial cells compared with that of the HOSE1 EVs (Fig. 5h). Next, to determine whether *MMP1* mRNA in EVs was involved in the destructive phenotypes of ES-2 EVs, we established ES-2 *MMP1* KD cells (Supplementary Fig. 10d,e). The EVs were isolated from ES-2 *MMP1* KD cells and control cells, and applied to HMPCs and caspase activity was assessed. As a result, the caspase activity in mesothelial cells treated with the EVs from ES-2 *MMP1* KD cells was significantly decreased (Fig. 5i). Furthermore, the caspase activity in MeT-5A cells transiently transfected with a *MMP1*-overexpressing vector was significantly increased (Fig. 5j). Taken together, these findings suggest that *MMP1* mRNAs were transferred to mesothelial cells via EVs and increased *MMP1* mRNA in mesothelial cells, thereby promoting apoptosis, although we cannot eliminate the possibility that ES-2 EVs stimulated *de novo MMP1* expression. Thus, our data indicate that *MMP1* mRNAs in EVs are key molecules in the destructive phenotypes in mesothelial cells.

**EVs carrying *MMP1* mRNA in patient-derived ascites**. Given that the EVs carrying *MMP1* mRNAs contributed to peritoneal dissemination in ovarian cancer, the *MMP1* expression levels could be a useful and critical marker of prognosis in ovarian

cancer patients. To test this hypothesis, we first examined the expression of *MMP1* in ovarian cancer, to determine its clinical relevance. Approximately 30% of ovarian cancer tissues expressed high levels of *MMP1* in 594 samples from The Cancer Genome Atlas database (Supplementary Fig. 11a). Furthermore, using a public database, a Kaplan–Meier analysis of ovarian cancer patients indicated that *MMP1* was a significant prognostic factor in 1,582 patients (hazard ratio (HR), 1.24; log-rank test, $P = 0.013$; Fig. 6a, left panel). Notably, it was a very strong prognostic factor, especially in 74 stage l patients (HR, $3.6 \times 10^8$; log-rank test, $P = 0.014$; Fig. 6a, right panel). These data supported the hypothesis that *MMP1* is an important gene in ovarian cancer malignancy.

As the EVs reflected the nature of the parental cells to some extent[27], we next determined whether ascites derived from ovarian cancer patients also contained EVs with abundant *MMP1* mRNA (Supplementary Fig. 11b). To verify this hypothesis, we isolated EVs from the ascites using ultracentrifugation and analysed them by phase-contrast electron microscopy and a nanoparticle tracking system, which revealed that the EVs in ascites had a typical EV structure and size of approximately 130 nm (Fig. 6b). Then, we measured the *MMP1* mRNA in EVs isolated from 60 patients' ascites by qRT–PCR (Fig. 6c, Supplementary Fig. 11c and Supplementary Table 2). As a result, a broad range of *MMP1* expression was detected across the samples, including control, low potential malignancy and cancer (Fig. 6c, left), and the percentage of *MMP1*-high population in cancer patients (27%) was comparable to that in The Cancer Genome Atlas database (Supplementary Fig. 11a). We initially assumed that the *MMP1* expression levels would increase according to the cancer stage; however, there were no statistically significant differences in the amount of *MMP1* mRNA in the EVs according to stage (early or advanced; Fig. 6c, middle panels) or histopathology (Supplementary Fig. 11c), thereby implying that the *MMP1* in ascites might be used as a risk indicator of peritoneal dissemination, even in the early stage. This possibility is consistent with the results shown in Fig. 6c (left graph). We also found higher amounts of *MMP1* mRNA in patients before chemotherapy than after chemotherapy (Fig. 6c, right panel), thus suggesting that chemotherapy reduced the secretion of malignant EVs from cancer cells, or that the cancer volume simply decreased. Collectively, our findings revealed that the EVs containing *MMP1* mRNA are present in patient-derived ascites and might reflect the status of the patients.

**EVs from patients carrying *MMP1* mRNA induce apoptosis**. To further investigate whether the ascites-derived EVs carrying *MMP1* mRNA also produce a destructive phenotype in mesothelial

---

**Figure 5 | *MMP1* mRNA in EVs is a key molecule for the destructive phenotypes in mesothelial cells.** (**a**) Schematic protocol for the gene expression analysis in mesothelial cells that were treated with EVs. (**b**) A heat map showing 482 differentially expressed genes (time point: 48 h, cancer EVs vs control EVs, change >1.5-fold and $P < 0.05$) in MeT-5A cells treated with EVs. (**c**) GSEA of the mesothelial cells treated with ES-2 EVs versus those with HOSE1 EVs, highlighting the destructive phenotypes. NES: a normalized enrichment score. The $P$-values in the graphs were calculated by GSEA analysis. (**d**) A heat map showing the expression levels of 27 genes (time point: 48 h, change >3-fold and $P < 0.05$) comparing mesothelial cells treated with ES-2 EVs with those treated with HOSE1 EVs. The lower panel shows the gene lists (top 20) of the above heat maps. (**e**) Schematic protocol for investigating the function of *MMP1* mRNA in EVs. This protocol was applied in **f,g**. (**f**) Immunoblot analysis of MMP1 and β-actin. The intensity of the bands (supernatant) was measured by LAS-4000. The intensity of backgrounds was subtracted in all samples. (**g**) MMP1 activity assay in the cell culture supernatant. The activity was measured using an enzyme-linked immunosorbent assay (ELISA) for MMP1. The concentration of active MMP1 protein is shown. Error bars represent s.d., Student's $t$-test ($^*P < 0.01$). (**h**) Caspase activity in Met-5A cells and HPMCs treated with EVs by using a Caspase-Glo 3/7 assay. The activity of both mesothelial cells treated with PBS was subtracted from those of ES-2 EV and HOSE1 EV. Error bars represent s.d. Student's $t$-test ($^*P < 0.01$). (**i**) Caspase activity in Met-5A cells and HPMCs treated with EVs derived from the *MMP1* shRNA-expressing ES-2 cells (KD) or control (NC, negative control). Error bars represent s.d. Student's $t$-test ($^*P < 0.01$). (**j**) Caspase activity in MeT-5A cells transfected with a *MMP1*-overexpressing vector (O.E.) or control (N.C.). Error bars represent s.d. Student's $t$-test ($^*P < 0.01$). The data are representative of at least three independent experiments each (**f–j**).

cells, similar to ES-2 EVs, the EVs from six patients were selected based on the amount of *MMP1* mRNAs and applied to HPMCs (Fig. 6d,e). Interestingly, the caspase activities in HPMCs treated with the patient-derived EVs with high amounts of *MMP1* mRNA

(cancer 1 and cancer 2) were significantly increased and the morphology of the HPMCs was altered in a similar manner to the ES-2 EV treatment (Fig. 6f). These findings suggest that the EVs carrying *MMP1* mRNA from patient-derived ascites also induce

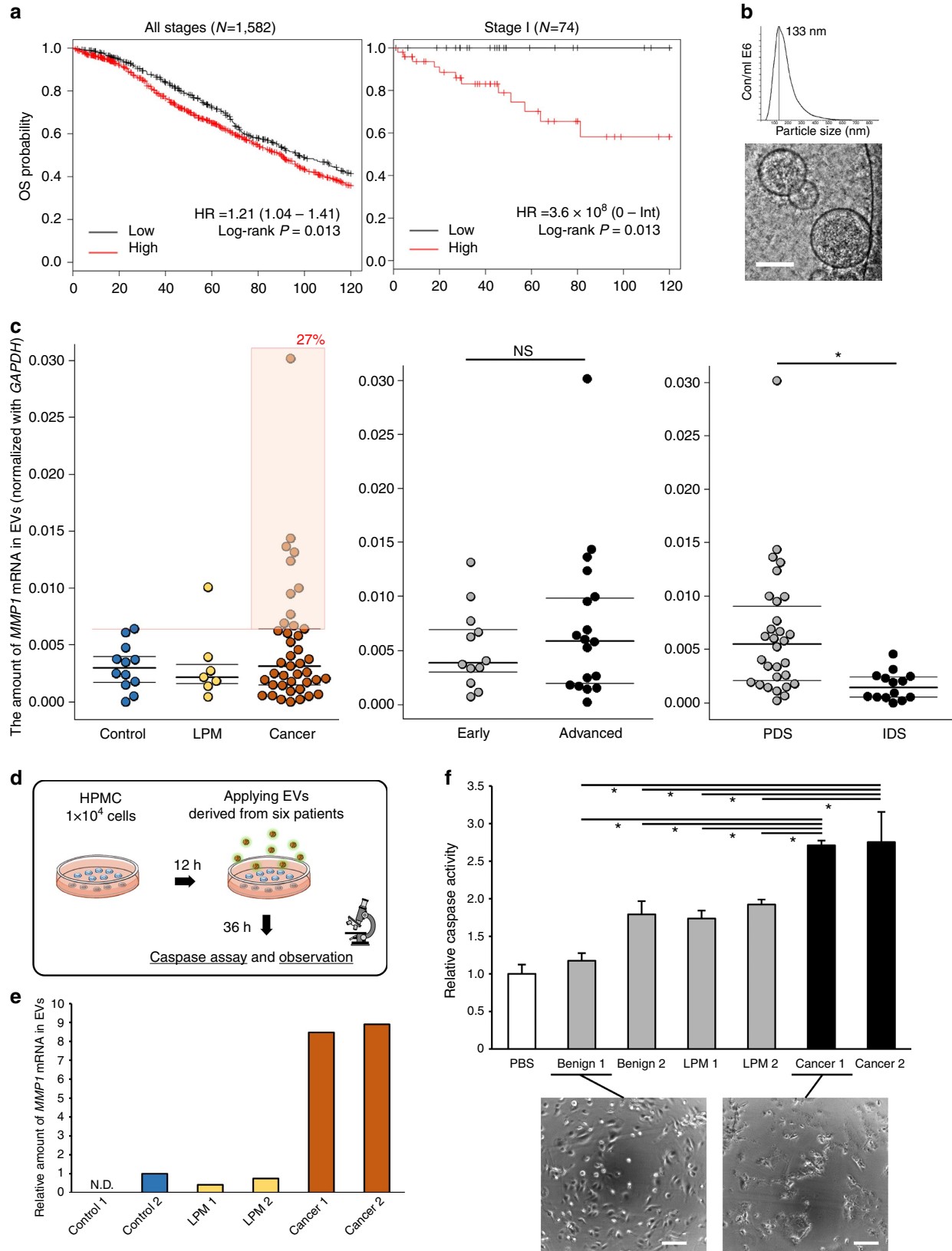

apoptosis in mesothelial cells, presumably leading to the initiation of peritoneal dissemination in the patients.

## Discussion

A number of reports have demonstrated that EVs mediate cancer metastasis by the horizontal transfer of bioactive molecules to recipient cells[7–11,20,28,29]. However, the detailed molecular and cellular mechanisms underlying EV-mediated metastasis have not been reported, especially in peritoneal dissemination, and this metastasis is responsible for the high morbidity and mortality observed in both ovarian cancer and gastrointestinal cancer[30,31]. In the present study, we identified the EVs that act as major mediators of peritoneal dissemination.

To understand the nature of ovarian cancer, it is necessary to consider its histological subtypes, because this cancer is not a single disease. Epithelial ovarian cancer consists of four major histological subgroups: serous, clear-cell, endometrioid and mucinous, which each have distinct molecular and pathological characteristics. In addition, serous and clear-cell types have the worst prognoses[32,33]. For these subtypes, the cells of origin for the ovarian cancer cells are derived from non-ovarian tissues, that is, the origin of the high-grade serous type is a distal fallopian tube, and that of the clear-cell type is an endometrial cyst, which is a common benign gynaecological tumour[34–38]. In this study, we prepared two histological subtypes of ovarian cancer cell lines (serous types A2780 and SKOV3 cells; clear-cell types ES-2 and RMG-1 cells). Then, the proportions of tumour progression of peritoneal dissemination were assessed by treating the cells with malignant EVs from ES-2 cells. Although this design is imperfect, it is sufficient to demonstrate the effect of malignant EVs on mesothelial exfoliation, which in turn promotes peritoneal dissemination. Moreover, many cancer cell lines do not reflect the histotypical characteristics of the original tumour tissues[39–41]. Peritoneal dissemination is a common and critical problem for patients, regardless of subtype.

Unexpectedly, we identified *MMP1* mRNA in EVs derived from a highly metastatic cancer cell line, ES-2 and patient ascites, and found that it was associated with the malignant phenotypes. It remains unknown why and how cancer cells package *MMP1* mRNA in EVs; nevertheless, it is conceivable that the use of EVs may offer an effective strategy for cancer progression. One possible explanation is that the nucleic acids in EVs are protected by RNase in body fluids and can remain intact and functional over a long period of time. Compared with miRNAs, mRNAs are fragile because of their longer size; therefore, protection by EVs might be more essential for mRNA stability in body fluids. mRNA stability and length can contribute to the efficient transfer of bioactive molecules to receiving cells and their rapid

accumulation effectively influences their phenotypes. The mRNA was translated in mesothelial cells (Fig. 5f). However, the difference of MMP1 protein level in cell lysates was not observed, because MMP1 is a type of secretory protein that is often analysed in cell culture supernatants[42,43]. In addition, the difference of protein level was not the same as the results of mRNA (Supplementary Fig. 9), presumably because the increase of *MMP1* mRNA in mesothelial cells was caused by the direct transfer via EVs and the part of the mRNA was translated.

Historically, MMPs have long been studied in cancer biology and the accumulated evidence has revealed that MMPs are related to cancer progression and prognosis[44–46]. MMPs mediate extracellular matrix and basement-membrane degradation during cancer progression in all stages, thereby contributing to the re-development of the surroundings and future metastatic sites[47,48]. Unfortunately, the current MMP research has stagnated because of disappointing clinical trial results[49]. In this study, we demonstrated that *MMP1* upregulation in mesothelial cells induces apoptosis. In general, MMPs have both apoptotic and anti-apoptotic functions[47,50,51]. Several reports have suggested that the function of MMPs in cancer cells is to mediate an escape from apoptosis[47,52,53]. However, especially in recipient cells, the overexpression of MMPs induces apoptosis[54–57]. Furthermore, MMP1 has been implicated to lead apoptosis in neurons and myocytes[58–60]. Although we have yet to identify the molecular mechanisms by which *MMP1* mRNA induces apoptosis in mesothelial cells, some reports have addressed how to induce apoptosis, for example, MMP1 decreases Akt activity and degrades laminin or releases Fas-ligand, resulting in activation of caspase[55–57,60]. MMPs are worth reinvestigating and may have renewed clinical significance, because they may even become a promising target for improving therapeutic outcomes from novel aspects, such as EVs related pathways.

As expected, EVs with high concentrations of *MMP1* mRNAs were detected in the ascites. Thus, *MMP1* in EVs could be a prognostic biomarker, because the *MMP1* expression level in tumour tissues is strongly related to prognosis in stage l ovarian cancer patients. Theoretically, analysis of the EVs carrying *MMP1* in the ascites from early-stage patients could predict their prognosis in the future and may contribute to the improvement of these outcomes through early diagnosis of metastasis and recurrence. However, one of the limitations to our analysis of the clinical samples was that we did not examine the patient treatment history, because the clinical samples were collected in recent years. Future studies will determine whether EV-associated *MMP1* mRNA in ascites can be a prognostic indicator. In ovarian cancer therapy, patients at all stages usually undergo surgery to remove as much cancer tissue as possible and to provide an accurate diagnosis[61]. If we were able to effectively inhibit the

**Figure 6 | EVs carrying *MMP1* mRNA are present in patient-derived ascites and the EVs also induce apoptosis in mesothelial cells *in vitro*.** (**a**) Kaplan–Meier analysis for the probability of overall survival in ovarian cancer patients according to the expression of *MMP1*. HRs and *P*-values (log rank *P*) are shown for each survival analysis. The left curve indicates the outcome of the patients in all stages, and the right one is from stage l patients. Kaplan–Meier curves were generated using the KMplot software from a database of public microarray data sets (http://kmplot.com/analysis). (**b**) Characterization of the EVs from patients' ascites. Nanoparticle tracking analyses showing the particle size of the EVs. Phase-contrast electron microscopy was used to visualize the EVs. Scale bar, 100 nm. (**c**) Distribution of *MMP1* mRNA in patient ascites-derived EVs. The left plot shows the amount of *MMP1* mRNA among control (benign diseases), LPM (low potential malignancy) and cancer patients. The orange box indicates high-expressing cancer patients compared with the controls. $n = 12$ control, 7 LPMs and 41 cancer. The middle plot shows the comparison between early-stage and advanced-stage patients. $n = 12$ early and 18 advanced. The right plot shows the comparison between patients who underwent PDS (primary debulking surgery) and IDS (intermediate debulking surgery). All patients who underwent IDS were treated with neoadjuvant chemotherapy. $n = 30$ PDS and 14 IDS. The amount of *MMP1* mRNA in EVs was normalized to *GAPDH*. Student's *t*-test (*$P < 0.05$ and NS, no significance). (**d**) Schematic protocol to investigate the effect of patient-derived EVs on mesothelial cells. The same number of EV particles was added to the HPMCs. (**e**) The amount of *MMP1* mRNA in EVs among six selected patients. Two patients were selected whose ascites-derived EVs were enriched in *MMP1* mRNA. The amount of *MMP1* mRNA in EVs was normalized to *GAPDH*. (**f**) Caspase activity in mesothelial cells treated with EVs from patients' ascites measured by the Caspase-Glo 3/7 assay. Error bars represent s.d., Dunnett's *t*-test (*$P < 0.01$). Representative microscopic images of mesothelial cells treated with two types of EVs. Scale bars, 100 μm.

function of malignant EVs during surgery by administration of an inhibitor in the peritoneal cavity, this treatment could prevent or reduce future metastasis. As our recent report, we disrupted cancer-derived EVs by therapeutic antibody administration in a metastatic cancer mouse model and drastically suppressed metastatic cells[62].

Overall, our findings clearly demonstrate that high-metastatic cancer cells secreted EVs carrying *MMP1* mRNAs, which effectively damaged mesothelial cells, promoting peritoneal dissemination. EV research is now attracting more attention and it may soon reach the next step, including clinical applications for diagnosis or therapeutics. We believe that these findings can lead to an improvement of clinical outcomes for ovarian cancer patients.

## Methods

**Cell culture.** Human ovarian cancer cell lines (ES-2 and SKOV3) and a normal human mesothelial cell line (MeT-5A) were purchased from the American Type Culture Collection. A human ovarian cancer cell line (A2780) was purchased from the European Collection of Cell Cultures. Additional human ovarian cancer cell lines (RMG-1 and SKOV3-Luc) were purchased from the Japanese Collection of Research Bioresources cell bank. HOSE1 and HOSE2 are the immortalized cell lines, which were established from primary HOSE cells following infection with retroviruses expressing mutant Cdk4, cyclin D1 and human telomerase reverse transcriptase[24]. HPMCs were isolated from surgical specimens of omentum with the consent of each patient, as previously described[63]. Briefly, small pieces of omentum were surgically resected under sterile conditions and trypsinized at 37 °C for 30 min. The suspension was then passed through a 200 μm pore nylon mesh to remove the undigested fragments and centrifuged at 760 *g* for 5 min. The cells were cultured in RPMI-1640 medium supplemented with 10% heat-inactivated fetal bovine serum (FBS). In subsequent experiments, cells were used during the second or third passage after primary culture. The HPMCs were identified by immunostaining using mouse monoclonal antibodies against cytokeratin 19 and vimentin (Dako, Glostrup, Denmark).

ES-2, SKOV3, SKOV3-luc and MeT-5A cells were cultured in McCoy's 5A medium (Thermo Fisher Scientific) supplemented with FBS and a 1% antibiotic–antimycotic solution (AA, Invitrogen, USA) at 37 °C in 5% CO$_2$. A2780 cells and HMPCs were cultured in RPMI-1640 medium (Thermo Fisher Scientific) supplemented with 10% heat-inactivated FBS and 1% AA at 37 °C in 5% CO$_2$. RMG-1, HOSE1 and HOSE2 cells were cultured in DMEM medium/Ham's F-12 medium (Thermo Fisher Scientific) supplemented with 10% heat-inactivated FBS and 1% AA at 37 °C in 5% CO$_2$.

**EV purification and analysis.** The cells were washed with PBS and the culture medium was replaced with advanced DMEM medium (Thermo Fisher Scientific) for ES-2 cells, ES-2 *MMP1* KD cells and SKOV3 cells; advanced DMEM medium/Ham's F-12 medium (Thermo Fisher Scientific) for RMG-1 cells, HOSE1 cells and HOSE2 cells; or advanced RPMI-1640 medium (Thermo Fisher Scientific) for A2780 cells containing an AA and 2 mM L-glutamine (but not FBS). After incubation for 48 h, the conditioned medium (CM) was collected and centrifuged at 2,000 *g* for 10 min at 4 °C. To thoroughly remove cellular debris, the supernatant was filtered through a 0.22 μm filter (Millipore). The CM was then used for EV isolation.

To prepare EVs, CM or the ascites from patients was ultracentrifuged at 35,000 r.p.m. using a SW41Ti rotor for 70 min at 4 °C. The pellets were washed with PBS, ultracentrifuged at 35,000 r.p.m. using the SW41Ti rotor for 70 min at 4 °C and resuspended in PBS. The protein concentration of the putative EV fraction was determined using a Quant-iT Protein Assay with a Qubit 2.0 Fluorometer (Invitrogen). To determine the size distribution of the EVs, nanoparticle tracking analysis was carried out using the Nanosight system (NanoSight) on samples diluted 500- to 1,000-fold with PBS for analysis. The system focuses a laser beam through a suspension of the particles of interest. These are visualized by light scattering using a conventional optical microscope perpendicularly aligned to the beam axis, which collects light scattered from every particle in the field of view. A 60 s video recorded all events for further analysis by the nanoparticle tracking analysis software. The Brownian motion of each particle was tracked between frames to calculate its size using the Stokes–Einstein equation.

**PKH67-labelled EV transfer.** Purified EVs derived from ES-2, RMG-1 and HOSE1 cells were labelled with a PKH67 green fluorescence labelling kit (Sigma-Aldrich, MO, USA). EVs were incubated with 2 μM of PKH67 for 5 min and washed five times using a 100 kDa filter (Microcon YM-100, Millipore) to remove excess dye. PKH67-labelled EVs were used to assess EV uptake *in vitro* and *in vivo*.

**Electron microscopy.** The isolated EVs were visualized using a phase-contrast transmission electron microscope (Terabase Inc., Okazaki, Japan) that can generate high-contrast images of the nanostructures of biological materials such as liposomes, viruses, bacteria and cells, without staining procedures that may damage the samples. The natural structure of the sample distributed in solution can be observed by preparing the sample using a rapid vitreous ice-embedding method and cryo-phase-contrast transmission electron microscopy.

To directly observe the peritoneal membrane of the mice, SEM was used. Dissected samples were fixed with 2% paraformaldehyde and 2% glutaraldehyde in 0.1 M cacodylate buffer (pH 7.4) at 4 °C overnight. The samples were additionally fixed with 1% tannic acid in 0.1 M cacodylate buffer (pH 7.4) at 4 °C for 2 h. After fixation, the samples were washed four times with 0.1 M cacodylate buffer for 30 min and postfixed with 2% osmium tetroxide in 0.1 M cacodylate buffer at 4 °C for 3 h. Then, the samples were dehydrated in graded ethanol solutions (50, 70, 90 and 100%). The procedure was as follows: 50 and 70% for 30 min each at 4 °C, 90% for 30 min at room temperature and 4 changes of 100% for 30 min each at room temperature. After dehydration, the samples were continuously dehydrated with 100% ethanol at room temperature overnight. Next, the samples were substituted into *tert*-butyl alcohol for 1 h and were incubated at 4 °C. The frozen samples were vacuum dried. After drying, the samples were coated with a thin layer (30 nm) of osmium using an osmium plasma coater (NL-OPC80NS; Nippon Laser & Electronics Laboratory, Nagoya, Japan). Then, the samples were observed by SEM (JSM-6340F, JEOL Ltd, Tokyo, Japan) at an acceleration voltage of 5.0 kV. The damaged sites with no microvilli (Supplementary Fig. 6a, red circles indicated) were counted in six randomly selected fields and are expressed as the total number of the sites.

***In vivo* studies.** Animal experiments were performed in compliance with the guidelines of the Institute for Laboratory Animal Research, National Cancer Center Research Institute (Number: T14-013); 6–7 weeks old female CB-17/Icr-scid/scidJcl mice (CLEA, Tokyo, Japan) were used in the experiments. The orthotopic ovarian cancer mouse model was established by injecting ovarian cancer cells (1 × 10$^6$ cells in 50 μl of PBS) into the left ovary, which was pulled out from a small incision on left back (Fig. 1a). Four types of ovarian cancer cell lines (SKOV3, A2780, RMG-1 and ES-2) were transplanted. For *in vivo* imaging, the mice were administered 150 mg kg$^{-1}$ D-luciferin (Promega, Madison, WI) by intraperitoneal injection. Ten minutes later, photons in the whole bodies of the animals were measured by assessing bioluminescence with an IVIS Spectrum imaging system (Caliper Life Science, Hopkinton, MA). The data were analysed using LIVINGIMAGE 4.4 software (Caliper Life Science). Tumour development was monitored once a week by bioluminescent imaging. To quantify the tumour progression, the photon values were compared with the value at the time of killing, which was set to 1.0 (Fig. 1e). To evaluate the metastatic tumours at the time of killing, the photon values of the tumours in the peritoneal cavity were obtained after dissection of the primary tumour using the IVIS system (Fig. 2b and Supplementary Fig. 2a). The dissected tissues were promptly embedded in optimum cutting temperature compound (Tissue-Tek, SAKURA) and frozen in liquid nitrogen for section analysis. For observation with haematoxylin and eosin staining, the tissues were quickly fixed in 10% neutral buffered formalin and blocked in paraffin.

**RNA extraction and PCR analysis.** Total RNA was extracted from EVs, cultured cells or EVs derived from patient ascites using QIAzol and the miRNeasy Mini Kit (Qiagen, Hilden, Germany) according to the manufacturer's protocols. For qRT–PCR analysis, complementary DNA was generated from 1 μg of total RNA using a High Capacity cDNA Reverse Transcription Kit (Applied Biosystems). Real-time PCR was subsequently performed in triplicate with a 1:15 dilution of cDNA using Platinum SYBR Green qPCR SuperMix UDG (Invitrogen) on a StepOne Real-Time PCR System (Applied Biosciences). The data were collected and analysed using StepOne Software v2.3 (Applied Biosciences). All mRNA quantification data from cultured cells were normalized to the expression of glyceraldehydes 3-phosphate dehydrogenase (*GAPDH*) and all primer sequences are listed in Supplementary Table 3. To detect *MMP1* mRNA in EVs by RT–PCR, total RNA was extracted from ES-2 cells and ES-2 EVs as described above. To synthesize first-strand cDNA from these RNA samples, the SuperScript III First-Strand Synthesis System for RT–PCR (Invitrogen) was used according to the manufacturer's protocols. Then, the cDNA products were subjected to PCR using TaKaRa Ex Taq (TaKaRa) according to the manufacturer's protocols and using primers corresponding to the full-length *MMP1* cDNA (forward primer, 5′-GAT ATTGGAGCAGCAAGAGG-3′; reverse primer and 5′-CACCTTCTTTGGAC TCACAC-3′), which generates a 1,528 bp product. The resulting PCR products were analysed by 1.5% agarose gel electrophoresis and bands were visualized by ethidium bromide staining (Supplementary Fig. 10c).

**Microarray and bioinformatics.** To perform a mRNA microarray, 5 × 10$^4$ MeT-5A cells were seeded into 24-well plates and 12 h later the EVs from five types of cells (ES-2 cells, A2780 cells, SKOV3 cells, HOSE1 cells and HOSE2 cells) were added to MeT-5A cells at two different concentrations of EVs (10 μg or 30 μg per each well). Total mRNAs were extracted from the MeT-5A cells 12 or 48 h after treatment with the EVs.

Total RNA was amplified and labelled with Cy3 using a Low Input Quick Amp Labeling Kit, one colour (Agilent Technologies), following the manufacturer's instructions. In brief, 100 ng of total RNA was reverse-transcribed to double-stranded cDNA using a poly dT-T7 promoter primer. Primer, template RNA and quality-control transcripts of known concentration and quality were first denatured at 65 °C for 10 min and incubated for 2 h at 40 °C with 5 × first-strand buffer, 0.1 M dithiothreitol, 10 mM deoxynucleotide triphosphate mix and AffinityScript RNase Block Mix. The AffinityScript enzyme was inactivated at 70 °C for 15 min. cDNA products were then used as templates for in vitro transcription to generate fluorescent complementary RNA (cRNA). cDNA products were mixed with a transcription master mix in the presence of T7 RNA polymerase and Cy3-labelled CTP (cytidine 5′-triphosphate) and incubated at 40 °C for 2 h. Labelled cRNA was purified using RNeasy Mini Spin Columns (Qiagen) and eluted in 30 μl of nuclease-free water. After amplification and labelling, cRNA quantity and cyanine incorporation were determined using a NanoDrop ND-1000 spectrophotometer and an Agilent Bioanalyzer. For each hybridization, 0.60 μg of Cy3-labelled cRNA was fragmented and hybridized at 65 °C for 17 h to an Agilent SurePrint G3 Human GE v2 8x60K Microarray (design ID: 039494). After washing, the microarray chips were scanned using an Agilent DNA microarray scanner. Intensity values of each scanned feature were quantified using Agilent Feature Extraction software version 11.5.1.1, which performs background subtractions. We only used features that were flagged as no errors (detected flags) and excluded features that were not positive, not significant, not uniform, not above background, saturated and population outliers (compromised and not detected flags). Normalization was performed with Agilent GeneSpring version 13.1.1 (per chip: normalization to 75th percentile shift). There are a total of 50,599 probes on the Agilent SurePrint G3 Human GE v2 8x60K Microarray (design ID: 039494) without control probes. The altered transcripts were quantified using the comparative method. Raw and normalized microarray data are available in the Gene Expression Omnibus database (accession numbers GSE80125).

The intensity values were log$_2$-transformed and imported into the Partek Genomics Suite 6.6 (Partek Inc., Chesterfield, MO, USA). For gene expression analysis, a one-way analysis of variance was performed to identify differentially expressed genes. P-values and fold-change numbers were calculated for each analysis. Unsupervised clustering and heat map generation were performed with sorted datasets by Pearson's correlation on Ward's method with selected probe sets by Partek Genomics Suite 6.6. GSEA (www.broadinstitute.org/gsea) was performed to compare ES-2 EV-treated and HOSE1 EV-treated mesothelial cells.

**Caspase-3/7 activation assays.** A luminescent caspase-3/7 activation assay was performed on each sample according to the manufacturer's instructions (Caspase-Glo 3/7 assay, Promega). Briefly, mesothelial cells, MeT-5A cells and HMPCs were seeded in a white clear-bottom 96-well plate. Twelve hours later, 6 μg of EVs was added to each well and incubated for 48 h. Then, Caspase-Glo reagent was added and incubated for 1 h, and the enzymatic activity of caspase-3/7 was measured using a microplate reader (Gen5, BioTek Instruments, Winooski, VT, USA).

**Plasmids.** For luciferase-based reporter gene assays, pLucNeo was constructed by inserting a firefly luciferase gene derived from pGL3-control (Promega) into the pEYFP-1 vector (Clontech) at the BglII and AflII sites[19]. KD short hairpin RNA (shRNA) vectors for human nSMase2 (the two different target sequences: 1, 5′-CAACAAGTGTAACGACGAT-3′ and 2, 5′-GGAGATTTCAACTTTGATA-3′) and MMP1 were purchased from TaKaRa Bio. For MMP1 overexpression assays, pcDNA4-MMP1 was constructed as follow: full-length MMP1 was PCR-amplified from ES-2 EV-isolated total RNA. Three prime A-overhangs were added to the PCR products after 15 min of regular Taq polymerase treatment at 72 °C. The PCR products were cloned into pGEM-T Easy Vector (Promega). The pGEM-T Easy Vector carrying MMP1 was digested with the restriction enzymes KpnI and EcoRI. The fragments containing MMP1 amplified and the products were ligated into a pcDNA4/TO plasmid (ThermoFisher), generating pcDNA4-MMP1. The plasmids were verified by DNA sequencing.

**Transfection of plasmid DNA.** For establishment of stable cell lines, pLucNeo and the KD shRNA vector for human nSMase2 and MMP1 were used. Stable RMG-1, ES-2 and A2780 cell lines expressing firefly luciferase were generated by selection with 1 mg ml$^{-1}$ geneticin. Stable ES-2 and A2780 nSMase2-modified cell lines that expressed human nSMase2 shRNA were generated by selection with 1 μg ml$^{-1}$ puromycin. Stable ES-2 MMP1-modified cell lines that expressed human MMP1 shRNA were generated by selection with puromycin. RMG-1, ES-2 and A2780 cell lines were transfected with 0.5 μg of the vector at 90% confluency in 24-well plates using Lipofectamine LTX/Plus reagent according to the manufacturer's protocol. Twelve hours after transfection, the cells were replated in a 15 cm dish followed by a 2-week selection with the antibiotics. Ten surviving single colonies were picked from each transfectant and were then cultured for another 2 weeks. The cells expressing the highest levels of firefly luciferase or the lowest levels of nSMase2 and MMP1 among the transfectants were used as stable transfected cells. In the stable nSMase2 KD cells, the amount of EVs in the cell culture supernatant was also

verified and the cells with the lowest number of EVs were used as the stable transfected cells.

For MMP1 overexpression assays, transfection of the pcDNA4-MMP1 plasmid into MeT-5A cells was performed using Lipofectamine LTX/Plus reagent (Invitrogen). Cell numbers and amounts of plasmids for each transfection were determined according to the manufacturer's protocol. Total RNA was extracted at 48 h after transfection and caspase activation assays were performed.

**Immunoblot analysis.** Whole-cell lysates were prepared with Mammalian Protein Extract Reagent (M-PER; Thermo Scientific, Rockford, IL). After collecting the CM, cells in a 24-well culture plate were washed with PBS and 500 μl M-PER was then added. Whole-cell lysates were transferred to a 1.5 ml tube and sonicated. Proteins (cell lysate, EVs and cell culture supernatant) were loaded onto a Mini-PROTEAN TGX Gel (4–15%, Bio-Rad) and electrotransferred (100 v, 30 mA). The proteins were transferred to a polyvinylidene difluoride membrane (Millipore). After blocking in Blocking One (Nacalai Tesque, Kyoto, Japan), the membranes were incubated for 1 h at room temperature with primary antibodies, which included anti-CD63 (purified mouse anti-human CD63, H5C6, 556019, BD, dilution 1:200), anti-CD9 (ALB6, sc-59140, Santa Cruz Biotechnology Inc., dilution 1:200), anti-actin (clone C4, MAB1501, Millipore, dilution 1:1,000) and anti-MMP1 (EP1247Y, ab52631, Abcam, dilution 1:200). Secondary antibodies (horseradish peroxidase-linked anti-mouse IgG, NA931 or horseradish peroxidase-linked anti-rabbit IgG, NA934, GE Healthcare) were used at a dilution of 1:2,000. The membrane was then exposed to ImmunoStar LD (Wako, Osaka, Japan). Uncropped scans of the blts owere shown in Supplementary Fig. 12.

**Immunohistochemical staining.** Tissue samples were fixed with formalin and embedded in paraffin. Following dewaxing and rehydration, heat-induced epitope retrieval was performed by boiling the specimens in 1/200 diluted ImmunoSaver (Nissin EM, Tokyo, Japan) at 98 °C for 45 min. Endogenous peroxidase was inactivated by treating the specimens with 0.3% H$_2$O$_2$ in methanol at room temperature for 30 min. Then, the specimens were treated with 0.1% Triton X-100 for permeabilization. After treatment with a blocking reagent (Nacalai Tesque) at 4 °C for 30 min, the specimens were incubated with primary antibodies, anti-mouse mesothelin (295D, D053-3, MBL, dilution 1:100), at room temperature for 1 h or at 4 °C overnight. The sections were stained using ImPRESS IgG-peroxidase kits (Vector Labs) and a metal-enhanced DAB substrate kit (Life Technologies), according to the supplier's instructions. After counterstaining with haematoxylin, specimens were dehydrated and mounted. For actin staining, mesothelial cells were fixed with 4% paraformaldehyde for 10 min at room temperature. After washing with PBS, actin was stained with ActinGreen 488 Ready Probes Reagent (R37110, Molecular Probes). Then, after washing with PBS, the nuclei were counterstained with Hoechst 33342 dye (Dojindo). The stained cells were washed in PBS for observation.

**Quantification of microscopic images in vitro.** The number of the abnormal cells with spindle shapes (Fig. 4d) and the open area generated by treatment with EVs (Fig. 4e) were observed with a BZ-X700 microscope (Keyence, Japan) and analysed using the image analysis application for the BZ-X700 microscope (Keyence). To assess the shapes of the cells, abnormal shape was defined to be over the cutoff value of the aspect ratio, which was set as 0.8, and analysis was performed using the application software in 49 randomly selected fields and expressed as a relative number (Fig. 4d). The open area (μm$^2$) determined by the application software was analysed in six randomly selected fields and expressed as relative values (Fig. 4e).

**TUNEL assay.** Apoptosis of the mesothelial cells in the tissue samples from the mice treated with EVs was determined using the TUNEL technique. TUNEL assays were performed with the DeadEnd Colorimetric TUNEL System (Promega) according to the manufacturer's protocol.

**MMP1 activity assay.** MMP1 activity in the cell culture medium was measured using a SensoLyte Plus 520 MMP1 assay kit (Anaspec, Fremont, CA) according to the manufacturer's protocol. The cell culture medium of mesothelial cells treated with EVs was collected 48 h after changing the medium. The CM was used after centrifugation at 2,000 g for 10 min at 4 °C. The medium was incubated with a substrate that fluoresces when cleaved by MMP1. The fluorescence was measured by a microplate reader (Gen5, BioTek Instruments). The concentration of active MMP1 in the medium was determined using a calibration curve.

**Clinical ascites samples.** Collection and usage of human ascites from ovarian cancer patients (n = 48) and patients with benign diseases (n = 12) were approved by the Institutional Review Board at the National Cancer Center (number: 2014–164) and all materials were obtained with written informed consent. Ascites was collected during surgery or by puncturing the abdominal wall. In some cases, ascites can be collected after the addition of normal saline, referred to as washed ascites. After collection, it was immediately centrifuged at 2,000 g for 10 min at 4 °C. To thoroughly remove debris, the supernatant was filtered through a 0.45 μm

filter (Millipore) and kept at 80 °C until use. The ascites was then used for EV isolation, following the method of EV isolation described above.

To investigate the amount of *MMP1* mRNA in EVs by qRT–PCR, values were normalized to *GAPDH* mRNA, after identifying a correlation between the amount of *GAPDH* by qRT–PCR and RNA concentration measured by a bioanalyser (Agilent) (Supplementary Fig. 13a).

In general, protein concentration was used to quantify the amount of EVs, but we found that there was no correlation between the particle number and protein concentration among the EVs from patient ascites (Supplementary Fig. 13b). Thus, in *in vitro* experiments using the EVs from ascites (Fig. 6d–f), the same particle number of EVs ($9 \times 10^9$ particles per well) was used for the treatment of HMPCs.

**Kaplan–Meier analysis.** Kaplan–Meier analysis was performed using the KMplot software from a database of public microarray data sets (http://kmplot.com/ analysis). The results were collected from 1,582 ovarian cancer patients at all stages and 74 stage l patients. Kaplan–Meier plots were generated for the MMP1 probe (204475_at). To analyse the prognostic value of the probe, the samples were split into two groups according to the cutoff value generated by the software (all stage; 47, stage I; 38). HRs and *P*-values (log rank *P*) are shown for each survival analysis.

**Statistical analysis.** Unless otherwise stated, the data are presented as the mean ± s.d. and statistical significance was determined by a Student's *t*-test. In the graphs of dot plot, the bars indicated median and interquartile range, and statistical significance was determined by a Dunnett's test. $P < 0.05$ was considered statistically significant.

**Data availability.** The microarray data that support this study are available through the NCBI database under accession GSE80125. The gene expression data in Supplementary Fig. 11a and the Kaplan–Meier analysis in Fig. 6a referenced during the study are available in a public repository from the websites (http://kmplot.com/ and http://www.oncomine.org). All other relevant data are available within the article file or Supplementary Information, or available from the authors on reasonable request.

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

## Acknowledgements

We thank the members of the Molecular and Cellular Medicine laboratory for the helpful discussions. We also thank Nobuyoshi Kosaka for a critical review of this paper and suggestions. We are grateful to Tomoko Sumikura, Mayumi Kobayashi and Takashi Uehara for managing the clinical samples. This work was supported in part by a Grant-in-Aid from the Japan Science and Technology Agency (JST) through the Center of Open Innovation Network for Smart Health (COINS) initiated by the Council for Science; a Grant-in-Aid from the Basic Science and Platform Technology Program for Innovative Biological Medicine; a Grant-in-Aid from the Project for Development of Innovative Research on Cancer Therapeutics (P-Direct); Project for Cancer Research and Therapeutic Evolution (P-CREATE); the National Cancer Center Research and Development Fund (Core Facility; 26-A-3); and the 'Development of Diagnostic Technology for Detection of miRNA in Body Fluids' grant from the Japan Agency for Medical Research and Development (AMED).

## Author contributions

A.Y., Y. Yoshioka and T.O. designed the experimental approach. A.Y. and Y. Yoshioka performed the experiments and analysed the data. A.Y. and Y. Yamamoto performed the microarray analysis and analysed the results. A.Y. and T.O. wrote the manuscript, and Y. Yoshioka and Y. Yamamoto assisted. M.I., S.-i.I. and T.K. provided the patients' ascites samples. T.K. provided the HOSE1 and HOSE2 cell lines. F.T. supported technical procedure about animal experiments. H.K. and F.K. provided helpful discussions. The manuscript was finalized by T.O. with the assistance of all the authors.

## Additional information

**Competing financial interests:** The authors declare no competing financial interests.

**Publisher's note**: 
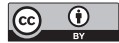

