## [Peer Review File · Nature Communications]

Reviewers' comments:

Reviewer #1 (Remarks to the Author):

This study attempts to demonstrate that extracellular vesicles (EVs) shed by aggressive ovarian cancer cells promote metastatic abilities of less aggressive cells. The authors propose this as a mechanism of peritoneal metastasis that is dependent on EV-induced increased expression of MMP1. They also suggest that EVs from metastatic cells induce apoptosis in mesothelial cells thus defeating the peritoneal mesothelium barrier. Finally, because MMP1 expression in ovarian cancer is found to correlate with histological parameters, the authors propose to use MMP1 in EVs in ascites as a potential prognostic marker.

The study is potentially interesting and presents some elements of novelty. Unfortunately a number of major issues decrease the significance. In particular, the rationale for most of the experiments is weak, the study is descriptive with results that are overall correlative and lack of mechanistic depth. The interpretation of the results is inflated.

Major issues:

It is confusing to hear about the animal experiment before the EV characterization is mentioned (and Figure 2a is referred to in the result section before Figure 1e).

The details regarding the cell lines used as EV donors for the EVs characterized here are missing, and it is not clear if the EVs are derived from ES-2 cells, from HOSE1, or both.

The results are not described clearly. For example the paragraph talking about an interaction between mesothelial cells and cancer EVs (Supplementary Fig. 3) is not very cohesive and doesn't really explain what type of interaction is going on. The text overall would benefit of extensive revision from a native speaker.

It is unclear whether the data showing that ES-2 cells, a highly and rapidly metastatic cell line, possessed the highest malignant potential, has been published or displayed elsewhere.

In the first paragraph the authors comment on the results regarding two of the 4 cell lines used and completely omit to talk about what they observe regarding the other two cell lines.

The rationale to study EVs from the ES-2 cell line and use as control EVs from a 5th cell line, which has not been previously tested, is unclear.

The rationale for using a differential centrifugation at 35,000 x g is not provided. Also, it is unclear why the NTA is used as a method to characterize these EV population considering that the pellets at 35,000 rpm might contain a significant portion of larger size EVs. Additionally, the EVs are further purified by using a centrifugation gradient. Furthermore, the method of EV injection in vivo is missing and the rationale for doing a certain number of injections in the peritoneum is not provided.

It is puzzling that MMP1 is the only mRNA out of 12 that is validated by PCR.

On the clinical side, it is unclear whether MMP1 could be used as an early indicator of disease or as a prognosticator.

The naMASE2 knock-down induces reduction in peritoneal metastasis in both more and less aggressive

cell lines, and the possibility that the metastasis reduction could be an off target effect of the knock down of the protein is not considered.

--

Reviewer #2 (Remarks to the Author):

In this study, the authors characterized the role of extracellular vesicles (EV) in peritoneal dissemination and metastasis in an orthotopic transplant mouse model of ovarian cancer. They found that these EV contains MMP1 mRNA, which was responsible to ruin the mesothelium barrier. In addition, expression of MMP1 showed clinical relevant in ovarian cancer patients. There are major concerns, and some conclusions are not solid. The work does not fit with Nature Communications.

1) To better understand the contribution of EV in cancer cell metastatic, each cells line should be compared for their metastatic ability after nSMase2 knockdown.

2) Is there any difference between the uptakes of ES-2 and HOSE1 EV by mesothelial cells? The relative percentage of abnormal MeT-5A cells after different EV treatment were ~1% compared to ~2%, which has very litter biological meaning.

3) Apoptotic analysis of mesothelial cells should be examined in mesothelial after high dose EV treatment in figure 4.

4) To understand the possible molecular mechanisms, the authors compared the global gene expression of mesothelial cells after different EV treatment. It will be useful if the authors can provide some detailed analysis of mesothelial cells treated with low and high metastatic EVs.

5) To further support the relationship between MMP1 in EV and metastasis, the expression of MMP1 in EV from each cell lines with different metastatic ability needed to be characterized.

How MMP1 induces apoptosis of mesothelial was not very well addressed in the study. It is quite surprising that this extracellular matrix metalloproteinase can be so heavily involved in apoptosis, given previous study has also shown it can protect lung alveolar epithelial cell from apoptosis.

Some minor issues:

Page 4, line 1, redundant space in the text.

Figure 2b, the picture didn't show any bioluminescent signal in the fifth mouse. In addition, the author didn't specify the details of the representative mice shown in figure2b. Ideally, four mice represent each group should be shown.

Figure3f. The TUNEL staining of controls should be shown. Quantification needed.

Figure 4e, the phase contrast photos show poor quality.

Figure 5f, the intensity of the bands need be quantified.

Page 12, line 3, a redundant comma

--

Reviewer #3 (Remarks to the Author):

The authors show extra-cellular vesicles (EVs) are secreted by ovarian cancer and normal epithelial cells; that these EVs are taken up by mesothelial cells and can damage these cells by changing the cellular morphology and inducing apoptosis. These changes promote peritoneal metastases. They show that the most profound effects are seen with EVs with high MMP1 levels and that high MMP1 tumour mRNA levels are associated with a poorer overall survival in women.

The concepts examined and results are highly novel and of great interest to cancer researchers. However there are some concerns that need to be addressed. The initial results are only shown for 1 cell line; additional controls are required for some experiments and statistical analysis does not always appear appropriate (comparisons appear to be made to ES-2 EVs rather than PBS). For these reasons it was often difficult to follow.

Comments

The implantation into the mouse ovary is not well explained. Are the cells injected into the body of the ovary or into the space between the ovary and bursa / membrane covering the mouse ovary? 50uL is a huge amount to inject at either site. The possibility of leakage is a real possibility that may confound the results. How was this controlled for?

An increase in metastasis is only shown in 1 cell line that is already metastatic (A2780) and requires repeated administration of EVs from a different cell line. Showing a change from non-metastatic to metastatic would have been more convincing. What is the concentration of EVs in ascites from the mice and how does this compare to the concentration in the mice treated every 2nd day with EVs from conditioned media? Assessment of mesothelial apoptosis in the presence of peritoneal metastases in the initial mouse models used would have helped to build the case for EV damage of mesothelium promoting peritoneal metastasis.

Do mesothelial cells secrete EVs?

Why was the development of ascites in the mice not used as factor? What % of mice developed ascites and was this related to EV or MMP1 levels?

While the authors show increases in caspase levels in mesothelial cells treated with ES-2 EVs relative to HOSE EVs, the images suggest the EVs induce EMT rather than apoptosis.

The presence of ascites does not occur in normal individuals. Please clarify the benign conditions from which ascites was collected.

Fig 1B - left: The dashed line appears to circumscribe the fat pad not just the ovary. Please clarify.

Fig 1d: were there significant differences in growth rates between the different cell lines?

Fig 2b: This needs a better explanation. Are the images from mice that received or did not receive EVs? It is stated that these are images at sacrifice on Day 21. When were the primary tumors removed (lower panel) to assess metastases?

Fig 2c: The statistical comparisons appear to be against ES-2 EVs rather than PBS (This also appears to be the case for many of the other figures). Please clarify or correct. The n's for each group need to be written correctly in the legend. The results for c and d appear to show medians not means as stated in Methods. Please clarify.

Fig 2d: ** represents a P-value greater than 0.05 and is therefore not significant as stated in Methods. This needs to be changed in the figure and Results section.

Fig 3: (c-f) This should include results for EVs from HOSE to determine whether these results are specific to EVs from malignant cells.

Fig 3d - change "marge" to "merge"

In Fig 4 it is suggested that ES-2 EVs damage the mesothelial MeT-5A cells. The images show a change from cobblestone to spindle shaped cells suggestive of EMT rather than apoptosis or cell damage.

Fig 5b: Gene expression is compared between mesothelial cells treated with different EVs. However, a mock treated control is not included to understand the basal levels of these genes.

Fig 5h: Caspase activity is compared between EVs from HOSE and ES-2 cells. A mock treated control needs to be included to understand the basal caspase activity of the cells and results presented as absolute values rather than relative values.

SIFig 2a needs a better explanation.

The title is unclear - perhaps change "toward" to "facilitating"

Minor editing of grammar is suggested.

Reviewer #1

We appreciate your critical comments, which have helped us improve the manuscript. All of the points that you raised have been addressed in the revision.

Comment_1: It is confusing to hear about the animal experiment before the EV characterization is mentioned (and Figure 2a is referred to in the result section before Figure 1e).

Response: The position of EV characterization was revised based on the reviewer's comment. It has been included in Supplementary Figure 1.

Comment_2: The details regarding the cell lines used as EV donors for the EVs characterized here are missing, and it is not clear if the EVs are derived from ES-2 cells, from HOSE1, or both.

Response: In an effort to avoid misunderstandings, the description has been included in the Figure legends of Supplementary Figure 1. We characterized the EVs derived from ES-2 cells, RMG-1 cells and HOSE1 cells by using nano particle tracking assays (Nanosight), western blotting analysis and electron microscopes. For all of the data, the origin of EVs was clearly displayed.

Comment_3: The results are not described clearly. For example the paragraph talking about an interaction between mesothelial cells and cancer EVs (Supplementary Fig. 3) is not very cohesive and doesn't really explain what type of interaction is going on. The text overall would benefit of extensive revision from a native speaker.

Response: We completely agree with Reviewer #1. As you noted, that part was confusing and somewhat ambiguous when we re-read it. The paragraph about the interaction between mesothelial cells and cancer EVs was substantially revised according to the reviewer's comment as shown in following texts, and the other sections were carefully reviewed. The entire manuscript was proofread by a native English speaker, and the certification is provided in the uploaded files.

“Here, we hypothesized that EVs derived from high metastatic cancer cells could have some effects on mesothelial cells because the peritoneum is the major obstacle that cancer cells must overcome for metastatic progression (Supplementary Fig. 4). Therefore, the effect

could be favorable for metastatic cancer cells in terms of demolishing the barrier, when the cells drop into the ascites, and when it creates metastatic sites. Thus, we evaluated those effects from cancer EVs to mesothelial cells.”

Comment_4: It is unclear whether the data showing that ES-2 cells, a highly and rapidly metastatic cell line, possessed the highest malignant potential, has been published or displayed elsewhere.

Response: ES-2 cells are frequently used ovarian cancer cell lines and have been used in many published studies ¹. Furthermore, they have been shown to have malignant and high-metastatic phenotypes ^{2,3}. In this work, we confirmed this high-metastatic ability for the first time by using an orthotopic mouse model.

Comment_5: In the first paragraph the authors comment on the results regarding two of the 4 cell lines used and completely omit to talk about what they observe regarding the other two cell lines.

Response: Commentary on the results of the other two cell lines was added in the first paragraph of the Results section as shown in the following text.

“A2780 cells and SKOV3 cells were also metastatic, but they required a longer time for peritoneal dissemination than ES-2 cells did.”

Comment_6: The rationale to study EVs from the ES-2 cell line and use as control EVs from a 5th cell line, which has not been previously tested, is unclear.

Response: Regarding the 5th cell line, you noted that it is the HOSE1 cell line. As detailed in the manuscript, the HOSE1 cell line is immortalized human ovarian surface epithelium, which has been previously established and characterized ⁴. The immortalized ovarian surface epithelium cells are the most commonly used control cells in ovarian cancer research ⁵⁻⁸. The HOSE1 cell was already tested in genome profiles and confirmed to be non-tumorigenic in previous work ⁴. Thus, we used the EVs from HOSE1 cells as control EVs.

Comment_7: (a) The rationale for using a differential centrifugation at 35,000 x g is not provided. Also, it is unclear why the NTA is used as a method to characterize these EV population considering that the pellets at 35,000 rpm might contain a significant portion of larger size EVs. Additionally, the EVs are further purified by using a centrifugation gradient. **(b)** Furthermore, the method of EV injection in vivo is missing and the rationale for doing a certain number of injections in the peritoneum is not provided.

Response: (a) We have routinely collected EVs by following standard methods in accordance to the position paper from International Society for Extracellular Vesicles (ISEV)⁹. The rationale for using the condition 35,000 rpm with a SW41Ti rotor is that this condition equates to approximately 100,000 x g in centrifugal force to the whole medium in tubes. As described in the Methods section “EV purification and analysis,” to remove cellular debris, the conditioned medium (CM) was centrifuged at 2,000 x g for 10 min at 4°C. To thoroughly eliminate larger-sized particles, it was then filtered through a 0.22- μ m filter (Millipore) before ultracentrifugation. The size of all EVs in this work was studied by nanoparticle tracking assay and electron microscopic analysis; the size was approximately 100 nm. These data indicated that larger EVs were successfully eliminated. We did not perform the gradient method, but our method is sufficient to determine that the EVs in our work were EV, as defined by ISEV. (b) We appreciate your helpful comments. The EV injection in vivo used the intraperitoneal method, which is included in the Figure Legends and the Methods section. Generally speaking, 1 - 30 μ g of EV had been used in previous EV experiment¹⁰⁻¹², however, using more than 30 μ g of EVs appeared excessive because of the number of original cells. We thought that it would be necessary to treat EVs continuously in mouse experiments because EVs are continuously released from cells. To avoid affecting the emerging growth of cancer cells, EVs were injected from day 3; considering the onset of metastasis, the injection period covered the first two weeks. These expectations and some pilot studies provided the rationale for the amount of EVs used in the in vivo experiments. To determine whether the amount of EV was excessive, a further in vivo experiment was performed by using a lower dose, as shown in Figure A. The results showed that the multiple injection method was effective in promoting metastasis; this result is included in Figure 2e, and the following texts was added in the revised manuscript.

“To determine the effect of ES-2 EVs on peritoneal dissemination in detail, we tested the dose-dependency of the EV injection (2 times, 4 times and 6 times) in the same experimental model (Fig. 2e and Supplementary Fig. 3c). Although a subtle increase in metastasis was observed after 4 injections, 6 injections obviously promoted metastasis, suggesting that the amount and timing of the EV injection are critical for the induction of peritoneal metastasis,

at least in our model. Collectively, these data clearly demonstrate that the EVs derived from metastatic ovarian cancer cells, particularly ES-2 cells, promote peritoneal dissemination *in vivo*.”

Figure. A Dose-dependency of ES-2 EVs in promoting peritoneal dissemination

(a) Schematic protocol for investigating the different doses of EVs in peritoneal dissemination. Orthotopic mouse models were established with A2780 cells. ES-2 EVs were injected i.p. beginning on day 3 and every other day thereafter. The names of the groups indicate how many times the EVs were injected. The mice were euthanized on day 21. (b) Distribution of

photon count in the dissected primary tumor (left) and the peritoneal metastatic tumors (right).
n = 5; 2 times, n = 5; 4 times, n = 5; 6 times. Dunnett's test (*p < 0.05 and NS = no significance).

Comment_8: It is puzzling that MMP1 is the only mRNA out of 12 that is validated by PCR.

Response: MMP1 was not the only mRNA of the 12 that was validated by PCR. Twelve genes were tested by PCR, and MMP1, G0S2, ADEM33 and SLC45A2 were validated by PCR (Supplementary Figure 9). However, the results were obtained from the condition using 30 µg and extracting RNA 48 h after treatment with EVs. In the PCR in various conditions, MMP1 was the most well-validated gene of the 12. For this reason, we decided to focus on MMP1.

Comment_9: On the clinical side, it is unclear whether MMP1 could be used as an early indicator of disease or as a prognosticator.

Response: In this work, we suggested that MMP1 could be used as a prognosticator, as described in the Discussion section (5th paragraph). As shown in Figure 6a, MMP1 could be a strong prognostic factor, especially in early-stage patients. However, as shown in Figure 6c, the patient stage was not related to the amount of MMP1 mRNA in EVs; there was no significant difference between controls and early-stage patients, as shown in Figure B. These factors highlight the potential of MMP1 as a prognosticator.

Figure. B MMP1 mRNA level in ascites EVs from patients

(a) Distribution of MMP1 mRNA in patient ascites-derived EVs among control (benign), early-stage and advanced-stage patients. The orange box indicates high-expressing cancer patients compared to the controls. The amount of MMP1 mRNA in EVs was normalized to GAPDH.

Comment_10: The nMASE2 knock-down induces reduction in peritoneal metastasis in both more and less aggressive cell lines, and the possibility that the metastasis reduction could be an off target effect of the knock down of the protein is not considered.

Response: To eliminate the possibility of an off-target effect, we established further cell lines by using other sequences of siRNA (TA0726 1C) for nSMase2 and performed additional in vivo experiments (Figure C). The nSMase2-knockdown A2780 and ES-2 cell lines were established, and the reduction in peritoneal metastasis in the mice with nSMase2-knockdown cells was confirmed. These results were combined with original data using another sequence of siRNA (TA0726 2B), as shown in Figure 2d and Supplementary Figures 2-3. Further description as the following texts was added in the Method section.

“Knockdown shRNA vectors for human nSMase2 (the two different target sequences: #1 – CAACAAGTGTAACGACGAT and #2 - GGAGATTTCAACTTTGATA)”

Figure. C Reduction in peritoneal metastasis by nSMase2 knockdown

(a) Characterization of nSMase2 knockdown (KD) cells. A2780 cells and ES-2 cells were transfected with the nSMase2 shRNA vector (TA0726 1C) or a negative control (NC) vector, and stable KD and NC cell lines were established. The number of EVs secreted from the same number of cells is shown in each left panel. The expression level of nSMase2 in each cell determined using qRT-PCR is shown in each right panel. Error bars represent s.d., Student's t-test (* $p < 0.01$). (b) Distribution of photon count in the dissected primary tumors (left) and the peritoneal metastatic tumors (right) using nSMase2 KD (TA0726 1C) cells. Orthotopic mouse models were established by injection with A2780 cells and ES-2 cells, and the cells were transfected with the nSMase2 shRNA vector or control vector. The mice with A2780 cells were euthanized at day 21 and those with ES-2 cells at day14. K.D.; knockdown of nSMase2, N.C.; negative control. Student's t-test (* $p < 0.05$ and NS = no significance).

Reviewer #2

We appreciate your critical comments, which have helped us improve the manuscript. All of the points that you raised have been addressed in the revision.

Comment_1: To better understand the contribution of EV in cancer cell metastatic, each cells line should be compared for their metastatic ability after nSMase2 knockdown.

Response: We appreciate your helpful comments. In this work, we used two cell lines, ES-2 and A2780, for the experiments on nSMase2 knockdown. We confirmed that RMG-1 cells had non-metastatic ability (Figure 1c). Therefore, it was impossible to use RMG-1 cells for those experiments. We thought that the remaining cell line, SKOV3, was not ideal for this experiment because it took over 2 months to produce metastatic tumors. Therefore, we established stable cells lines by using both potential cell lines and performed animal experiments to confirm the contribution of EV to cancer cell metastasis. Furthermore, we investigated the off-target effect according to the Reviewer's comment (Fig C). Taken together, the evidence of contribution was fully confirmed by our results. In this study, we focused not on the amount of EVs but on the properties of the EVs derived from high metastatic cancer cells. As shown in Figure 2c, the EVs from non-metastatic cancer cells did not promote metastasis. Thus, we found and suggested that the characteristics of EVs could be involved in metastatic phenotypes.

Figure. C Reduction in peritoneal metastasis by nSMase2 knockdown

(a) Characterization of nSMase2 knockdown (KD) cells. A2780 cells and ES-2 cells were transfected with the nSMase2 shRNA vector (TA0726 1C) or a negative control (NC) vector, and stable KD and NC cell lines were established. The number of EVs secreted from the same number of cells is shown in each left panel. The expression level of nSMase2 in each cell determined using qRT-PCR is shown in each right panel. Error bars represent s.d., Student's t-test (* $p < 0.01$). (b) Distribution of photon count in the dissected primary tumors (left) and the peritoneal metastatic tumors (right) using nSMase2 KD (TA0726 1C) cells. Orthotopic mouse models were established by injection with A2780 cells and ES-2 cells, and the cells were transfected with the nSMase2 shRNA vector or control vector. The mice with A2780 cells were euthanized at day 21 and those with ES-2 cells at day 14. K.D.; knockdown of nSMase2, N.C.; negative control. Student's t-test (* $p < 0.05$ and NS = no significance).

Comment_2: Is there any difference between the uptakes of ES-2 and HOSE1 EV by mesothelial cells? The relative percentage of abnormal MeT-5A cells after different EV treatment were ~1% compared to ~2%, which has very little biological meaning.

Response: The results of Supplementary Figure 7 concerning the uptake of EVs are quantified in Figure D. As shown in the results, there was no difference among each EV. Those data are now included in Supplementary Figure 7.

We apologize for the confusing data. The method for showing the difference in abnormal cells in Figure 4d is a relative number. In fact, the number of abnormal cells in MeT-5A treated

with ES-2 averaged approximately 18 cells in each view, and 49 views were analyzed. On the other hand, that of HOSE1 and PBS was approximately 10 cells. The percentage of abnormal cells in each view was altered from approximately 45% to 81%. The value was normalized to that of PBS, and this was significantly different between ES-2 EVs and HOSE1 EVs. The figure legend and the labels of the vertical axis were revised to further avoid misunderstanding.

Figure. D Quantification of uptake of various EVs by mesothelial cells.

The uptake of EVs by mesothelial cells (MeT5A and HPMC), as shown in Supplementary Figure 7, was quantified using the image analysis application for the BZ-X700 (Keyence, Japan). EV areas were normalized with the number of nuclei in 5 randomly selected fields and expressed as relative values.

Comment_3: Apoptotic analysis of mesothelial cells should be examined in mesothelial after high dose EV treatment in figure 4.

Response: The analysis was performed and shown in Figure 5h. The results of morphological analysis are shown in Figure 4d-e, and those of the apoptotic analysis are in Figure 5h.

Comment_4: To understand the possible molecular mechanisms, the authors compared the global gene expression of mesothelial cells after different EV treatment. It will be useful if the authors can provide some detailed analysis of mesothelial cells treated with low and high metastatic EVs.

Response: In this work, high metastatic EVs referred to the EVs from ES-2 cells, whereas the relatively low metastatic EVs were from A2780 and SKOV3 cells. The comparison of the gene expression in mesothelial cells treated with different metastatic EVs is shown in Figure E, which indicates that MMP1 is included in the high-expressing area in the mesothelial cells treated with ES-2 EVs. These data have been included in Supplementary Figure 8, and the following tests were added in the revised manuscript.

“To further investigate the relevance of MMP1 in mesothelial cells treated with various EVs, an additional analysis was performed using the data from a microarray analysis (Supplementary Fig. 8b-e). When we looked at gene expression change in mesothelial cells treated with low or high metastatic EVs, MMP1 was also contained in the population of up-regulated genes with high metastatic EVs (Supplementary Fig. 8b and c). Furthermore, MMP1 is one of the specifically up-regulated genes in mesothelial cells treated with highly metastatic EVs, ES-2 EVs (Supplementary Fig. 8d and e).”

b

c

d

Figure. E Gene expression analysis in the mesothelial cells treated with different EVs.

(a) A heat map showing 524 differentially expressed genes (time point: 48 h, ES-2 EVs vs A2780 EVs, change > 2-fold and $p < 0.05$) in MeT-5A cells treated with the EVs. (b) A heat map showing 367 differentially expressed genes (time point: 48 h, ES-2 EVs vs SKOV3 EVs, change > 2-fold and $p < 0.05$) in MeT-5A cells treated with the EVs. (d) Investigation of the specific gene in the mesothelial cells treated with ES-2 EV as shown in the Venn diagram (change > 2-fold and $p < 0.05$). (e) Investigation of the 6 specifically up-regulated genes in the mesothelial cells treated with ES-2 EV.

Comment_5: To further support the relationship between MMP1 in EV and metastasis, the expression of MMP1 in EV from each cell lines with different metastatic ability needed to be characterized.

How MMP1 induces apoptosis of mesothelial was not very well addressed in the study. It is quite surprising that this extracellular matrix metalloproteinase can be so heavily involved in apoptosis, given previous study has also shown it can protect lung alveolar epithelial cell from apoptosis.

Response: The amount of MMP1 mRNA in EV from each cell line was measured by qRT-PCR (Fig. F). The MMP1 levels in ES-2 EVs were the highest (much higher than others), followed by A2780 cells (moderate metastatic ability).

We were also surprised by the relationship between MMP1 and apoptosis, but our data clearly indicated that MMP1 induced apoptosis in mesothelial cells. As you noted in comment ¹³, MMP1-transfected epithelial cells avoided apoptosis by treatment with staurosporine or bleomycin, but the authors did not elucidate the specific pathway for anti-apoptosis. The function of MMPs for apoptosis remains elusive; several reports of MMPs-apoptosis exist ^{14,15}. Furthermore, the response is different depending on cell types, and MMP1 has been implicated to lead to apoptosis in neurons and myocytes ¹⁶⁻¹⁸. The mechanism by which ES-2 EVs induce apoptosis might be that MMP1 decreases Akt activity, degrade laminin or release FASL, resulting in the activation of caspase ¹⁸⁻²¹. In this revised version, we also confirmed that it did not relate to EMT phenotypes. A discussion of these points has been added to the Discussion section.

“In general, MMPs have both apoptotic and anti-apoptotic functions ^{48,49 45}. Several reports have suggested that the function of MMPs in cancer cells is to mediate an escape from apoptosis ^{45,50,51}. However, especially in recipient cells, the overexpression of MMPs induces

apoptosis⁵²⁻⁵⁵. Furthermore, MMP1 has been implicated to lead apoptosis in neurons and myocytes⁵⁶⁻⁵⁸. Although we have yet to identify the molecular mechanisms by which MMP1 mRNA induces apoptosis in mesothelial cells, some reports have addressed how to induce apoptosis, for example, MMP1 decreases Akt activity and degrades laminin or releases FASL, resulting in activation of caspase^{53-55,58}. MMPs are worth reinvestigating and may have renewed clinical significance because they may even become a promising target for improving therapeutic outcomes from novel aspects, such as EVs related pathways.”

Figure. F MMP1 mRNA levels in various EVs

Each RNA sample was extracted from 5 μg of EVs, and the amount of MMP1 mRNA in EVs was measured by qRT-PCR.

Comment_6: Page 4, line 1, redundant space in the text.

Response: The part which was pointed out in page 4 was revised according to the reviewer's comment.

Comment_7: Figure 2b, the picture didn't show any bioluminescent signal in the fifth mouse. In addition, the author didn't specify the details of the representative mice shown in figure 2b. Ideally, four mice represent each group should be shown.

Response: Actually the fifth mouse in the Figure had the tumors, but its signal was quite low. To avoid confusion, Figure 2b and its legend were substantially revised according to the reviewer's comment. Furthermore, three mice representing each group are shown in Figure G and included in Supplementary Figure 2.

Figure. G Representative bioluminescence images in *in vivo* experiments

(a) Representative bioluminescence images of orthotopic mouse models using A2780 cells at euthanasia. The upper images were captured from outside the mouse bodies using the *in vivo* imaging system. The middle images show dissected primary tumors. The lower images show the metastatic tumors in the peritoneal cavity. The primary tumors in these mice were already removed, as shown in the middle images. (b) Representative images for the results of experiments in Figure 2c. On the left, the images of dissected primary tumors are shown, and on the right, those of metastatic tumors are shown.

Comment_8: Figure 3f. The TUNEL staining of controls should be shown. Quantification needed.

Response: The control of the TUNEL staining assay was added as shown in Figure H, and it was added to Figure 3. Unfortunately, it was difficult to quantify the number of TUNEL-positive cells because there were no cells in the peritoneum treated with PBS or HOSE1 EVs.

Figure. H TUNEL staining assay in the abdominal wall of the mice treated with EVs

Representative microscopic images of TUNEL-stained slides from the abdominal wall and the omentum dissected from the mice treated with PBS, HOSE1 EVs and ES-2 EVs. Black arrowheads indicate positive cells. Scale bars, 100 μ m.

Comment_9: Figure 4e, the phase contrast photos show poor quality.

Response: The quality of the photos in Figure 4e might have been degraded when the manuscript was converted to a PDF. We have fair-quality photos, as shown in Figure I.

Figure. I Fair-quality of photos in Figure 4e

Analysis of the effect of EVs on HPMCs. EVs were added to HPMCs and observed after 48 h.

Comment_10: Figure 5f, the intensity of the bands need be quantified.

Response: The bands in Figure 5f are quantified in Figure J, and these results were included in Figure. 5f.

Figure. J The relative intensity of the bands in western blotting analysis.

The intensity of the bands in immunoblot analysis (Figure 5f) were measured with the analyzer in LAS-4000 (GE Healthcare, USA). The intensity of backgrounds was subtracted in all samples.

Comment_11: Page 12, line 3, a redundant comma

Response: The part in page 12 was revised according to the reviewer's comment.

Reviewer #3

We appreciate your critical comments, which have helped us improve the manuscript. All of the points that you raised have been addressed in the revision.

Comment_1: The implantation into the mouse ovary is not well explained. Are the cells injected into the body of the ovary or into the space between the ovary and bursa / membrane covering the mouse ovary? 50 μ L is a huge amount to inject at either site. The possibility of leakage is a real possibility that may confound the results. How was this controlled for?

Response: The explanation of the implantation was revised according to the Reviewer's suggestion and is further described in Figure legends for Figure 1a. The ovarian cancer cells were injected into the ovarian bursa; the amount of 50 μ L is not excessively large. As the photos in Figure K-a show, the bursa were enlarged after injection, and we confirmed that the bursa can hold a volume of 50 μ L without leakage. To test what would occur if the cancer cells leaked into the peritoneal cavity, we prepared a mouse with leakage. As shown in the in vivo imaging system depicted in Figure K-b, the tumors in the mouse with leakage were detected in multiple areas in the early phase. These data suggest that we can identify leakage through routine IVIS imaging. Furthermore, the IVIS images for the orthotopic mice with RMG-1 cells shown in Figure 1c show that they did not produce metastatic tumors within two months; in other words, there was no leakage. Taken together, it is possible to transplant the cells with no leakage and to find the mice with leakage.

Figure. K Investigation of leakage at transplantation of cancer cells

(a) The photos of left ovarian bursa before and after injection of the cancer cells on the day of the implantation. (b) The IVIS images at day 3 for the mice bearing ovarian cancer cells with leakage (+/-). The mouse with leakage (-) was injected with cells in the left ovarian bursa, whereas the mouse with leakage (+) was injected with cells IP through the incised hole on the left back.

Comment_2: (a) An increase in metastasis is only shown in 1 cell line that is already metastatic (A2780) and requires repeated administration of EVs from a different cell line. Showing a change from non-metastatic to metastatic would have been more convincing. (b) What is the concentration of EVs in ascites from the mice and how does this compare to the concentration in the mice treated every 2nd day with EVs from conditioned media? (c) Assessment of mesothelial apoptosis in the presence of peritoneal metastases in the initial mouse models used would have helped to build the case for EV damage of mesothelium promoting peritoneal metastasis.

Response: (a) We completely agree with your comments regarding the importance of using additional cell lines; a new experiment was performed by using other candidate cell lines, SKOV3 and RMG-1 cells, as shown in Figure L. The protocols were slightly modified according to the results of tumor progression speed shown in Figure 1c. The results indicated that metastasis in the mice transplanted by using SKOV3 was successfully promoted, whereas that of using RMG-1 was not changed after different EV injections. In this work, we investigated the ability of the EVs from high metastatic cancer cells to mesothelial cells and focused on how the EVs break the barrier of the peritoneal membrane. We did not look at the effect of the EVs on cancer cells, and we did not obtain direct evidence that the EVs phenotypically influenced the cancer cell themselves because there was no difference in primary tumor and the histology of tumors (primary tumor and metastatic tumor) did not obviously change (Figure L-c). SKOV3 was confirmed to be a moderately metastatic cell line, but RMG-1 was non-metastatic. We supposed that the EVs only promote metastasis but do not change the property of cancer cells from non-metastatic to metastatic. Therefore, we conclude that the results of those new experiments are reasonable and are consistent with the theory of this study. These data were included in Supplementary Figure. 2, and the following texts were added in the revised manuscript.

“To further examine the ability of EVs involved in peritoneal metastasis, the same experiments using the mice transplanted with either SKOV3 cells or RMG-1 cells were performed (Supplementary Fig. 2b). As a result, the metastasis in the mice with SKOV3 was

significantly promoted by ES-2 EVs, but metastasis in RMG-1 was not changed after successive EV injections. These data suggested that the high metastatic EVs could only promote metastasis but do not change the property of cancer cells from non-metastatic to metastatic.”

(b) It was difficult to measure the EV concentration from ascites in the mice because most of the mice in this study did not accumulate ascites. Furthermore, if we could collect the ascites, the amount is quite low, and it is insufficient to allow the collection of EVs from the ascites via ultracentrifugation. In this study, we used 5 – 50 μg / 500 μL of EVs for IP injection. The concentration of EVs was approximately 1×10^9 - 1×10^{10} particles/mL. The EV concentration in patients' ascites was quantified at approximately 4.5×10^{10} particles / mL. Therefore, the concentration of EVs used in this study was not in excess of the physiological levels.

(c) To assess mesothelial apoptosis in the presence of peritoneal metastases, TUNEL staining assay was newly performed. The peritoneal membrane was obtained from the mice that were orthotopically established, without EV injection, and had already metastasized, such as the mice shown in Figure 1c. A few TUNEL positive cells were found in the mice with ES-2 cells, but they were not found in other samples (Figure M-a). However, compared to distant areas, the structure of the peritoneal membrane proximal to metastatic tumors in the mice with A2780 and SKOV3 was changed (Figure M-b), such as after EV injection (Figure 3e and f). It was unclear whether the morphological change was caused by the EVs from cancer cells, but such a structural change might suggest progress toward apoptosis according to the images in Figure 3e and f. The concentration of EVs might be influenced because we demonstrated that ES-2 EVs contain a significant amount of MMP1, which is related to an apoptotic phenotype. Furthermore, although it might be natural to think that MMP1-high EVs could be a factor for peritoneal metastasis, other factors also induce metastasis without apoptosis.

Figure. L Investigation of further cell lines in additional *in vivo* experiments

(a) Upper: Schematic protocol for investigating the EVs in peritoneal dissemination. Orthotopic mouse models were established with RMG-1 cells. ES-2 EVs and HOSE1 EVs were injected i.p. from day 21 and on every other day thereafter, for a total of 6 times. The mice were euthanized on day 42. Lower: Distribution of photon count in the dissected primary tumor (left) and the peritoneal metastatic tumors (right). $n = 12$ (6; HOSE1 EV, 6; ES-2 EV). Student's t-test (* $p < 0.05$ and NS = no significance). (b) Upper: Schematic protocol for investigating the EVs in peritoneal dissemination. Orthotopic mouse models were established with SKOV3 cells. ES-2 EVs and HOSE1 EVs were injected i.p. from day 3 and on every other day thereafter, for a total of 6 times. The mice were euthanized on day 30. Lower:

Distribution of photon count in the dissected primary tumor (left) and the peritoneal metastatic tumors (right). $n = 12$ (6; HOSE1 EV, 6; ES-2 EV). Student's t-test ($*p < 0.05$ and NS = no significance). (c) Histological features of tumors in mouse model. The tumors were obtained from the orthotopic mouse model by using A2780. After metastasis, the tumors were dissected. HE staining. Scale bars, 50 μm .

a

Figure. M TUNEL staining assay in the mice treated with EVs

(a) Representative microscopic images of TUNEL-stained slides from the abdominal wall and the omentum dissected from the mice orthotopically transplanted by RMG-1 cells, SKOV3 cells, A2780 cells and ES-2 cells. Black arrowheads indicate positive cells. Scale bars, 100 μm .

(b) Representative microscopic images of TUNEL-stained slides from the abdominal wall dissected from the mice orthotopically transplanted by SKOV3 cells. Red box: proximal area to tumor; Blue box: distant area. Scale bars, 100 μm .

Comment_3: Do mesothelial cells secrete EVs?

Response: Yes, they do. We collected the EVs from mesothelial cells (MeT-5A cells), and characterized them in Figure N. The mesothelial cells also secreted a certain amount of EVs (17.2×10^8 particles / 1 ml of culture medium), and we confirmed that they contained almost no MMP1 mRNAs.

Figure. N Characterization of EVs from mesothelial cells

(a) Nanoparticle tracking analyses of the particle size of EVs. The vertical axis in the graph shows the number of EV particles ($\times 10^6$)/mL, and the horizontal axis indicates the particle size (nm). (b) The image of scattered light from EVs derived from mesothelial cells by using a nanoparticle tracking system (Nanosight). (c) Immunoblot analysis for EVs. CD9 and CD63 were detected in EVs. 50 ng / lane. (d) The amount of MMP1 mRNA in ES-2 EVs and MeT-5A EVs. All RNAs were extracted from 5 μ g of EVs, and the amount of MMP1 mRNA in EVs was measured by qRT-PCR.

Comment_4: Why was the development of ascites in the mice not used as factor? What % of mice developed ascites and was this related to EV or MMP1 levels?

Response: As you noted, ascites is a very important factor in assessing disease severity. Ovarian cancer is characterized by diffuse peritoneal carcinomatosis and, often, by large

volumes of ascites. However, ascites accumulation is not the cause of metastasis; instead, the effects of various factors such as disseminated tumor or indicators of patient deterioration, such as malnutrition, can lead to metastasis²²⁻²⁴. In this work, we focused on only the emergence of metastasis because we sought to investigate the function of EVs related to the onset of peritoneal dissemination. We confirmed that mice bearing a large tumor burden would accumulate ascites but that would take a amount of time (approximately two months); however, the time course in these in vivo experiments was too short to result in ascites formation. Thus, none of the mice in the experiment (Figure 1a) accumulated ascites. However, a few mice in the experiment shown in Figure 1a accumulated ascites, and we checked the level of MMP1 in EVs, as shown in Figure O. We found that the MMP1 level in the mouse bearing high-metastatic ovarian cancer cells was markedly higher than that found in the other mice.

Figure. O MMP1 mRNA level in mouse ascites EV

The mouse model was orthotopically established, and ascites were collected at euthanasia (each n = 2). EVs were purified by using the Total Exosome Isolation Kit (Invitrogen). The amount of MMP1 mRNA was measured by qRT-PCR.

Comment_5: While the authors show increases in caspase levels in mesothelial cells treated with ES-2 EVs relative to HOSE EVs, the images suggest the EVs induce EMT rather than apoptosis.

Response: We tested the induction of EMT by ES-2 EVs. The gene expression change in mesothelial cells treated with ES-2 EVs, HOSE1 EVs and PBS is shown in Figure P-a, and there was no evidence for EMT. Furthermore, we found that the expression of E-cadherin in mesothelial cells was quite low, and there were no differences in the results of the immunofluorostaining assay (Figure P-b).

Figure. P Investigation of the induction of EMT in mesothelial cells treated with EV

EMT related genes were investigated and are shown as a heat map. Gene expression in MeT-5A cells treated with the ES-2 EVs, HOSE 1 EVs and PBS was analyzed. (b) Immunofluorostaining assay in MeT-5A cells treated with the ES-2 EVs, HOSE 1 EVs and PBS. E-cadherin: (purified rabbit anti-human E-Cadherin, 24E10, 1:500, CST; green). Nuclei (Hoechst 33342 dye, Dojindo; blue).

Comment_6: The presence of ascites does not occur in normal individuals. Please clarify the benign conditions from which ascites was collected.

Response: Generally speaking, the peritoneal cavity in healthy people contains some fluid (between 10 mL and 50 mL) that serves as a lubricant. Thus, it is possible to collect ascites from every patient during surgery. However, in some cases, the amount is too low to obtain a sufficient amount for EV purification. In those cases, ascites can be collected after the addition of normal saline, referred to as washed ascites. For PCR analysis, the MMP1 level was normalized with the GAPDH level because there was a correlation between RNA concentration and the GAPDH level, as shown in Supplementary Figure 12a. The description of ascites collection in the Methods section was revised in response to the Reviewer's comments as shown in the following texts.

“Ascites was collected during surgery or by puncturing the abdominal wall. In some cases, ascites can be collected after the addition of normal saline, referred to as washed ascites. After collection, it was immediately centrifuged at 2,000 x g for 10 min at 4°C.”

Comment_7: Fig 1B - left: The dashed line appears to circumscribe the fat pad not just the ovary. Please clarify.

Response: We apologize for the confusing photos, but the fat pad did, in fact, easily adhere to the tumor. After removing the adhesions, the tumor with the uterus appeared as shown in Figure 1b – right. The description for the dashed line in Figure 1b - left was revised to clarify the meaning of the line in the legend as shown in the following text.

“The left photo shows metastatic tumors (black arrowheads) and the uterus with the primary left ovarian tumor (a black dotted line indicated the position of them), covered with fat pad.”

Comment_8: Fig 1d: were there significant differences in growth rates between the different cell lines?

Response: In the additional statistical analysis, there were significant differences at the 2 weeks time point by using ANOVA followed by the Tukey-Kramer test for ES-2 vs. other. Figure 1d and the methods section were revised to reflect these data. The following text was added in Figure Legend.

“There were significant differences at the 2 weeks time point by using ANOVA followed by the Tukey-Kramer test for ES-2 vs. other (* $p < 0.05$).”

Comment_9: Fig 2b: This needs a better explanation. Are the images from mice that received or did not receive EVs? It is stated that these are images at sacrifice on Day 21. When were the primary tumors removed (lower panel) to assess metastases?

Response: We apologize for the confusing images; the mice shown in the Figure received EVs. To avoid confusion, this section was substantially revised, as shown in Figure Q-a, and four mice representing each group are shown in Figure Q-b and have been included in Supplementary Figure 2a.

a

b

Figure. Q Representative bioluminescence images in *in vivo* experiments

(a) Representative bioluminescence images of orthotopic mouse models using A2780 cells at euthanasia. The upper images were captured from outside the mouse bodies using the *in vivo* imaging system. The middle images show dissected primary tumors. The lower images indicate the metastatic tumors in the peritoneal cavity. The primary tumors in these mice were already removed, as shown in the middle images. (b) Representative images for the results of experiments in Figure 2c. On the left, images of dissected primary tumors are shown, and on the right, those of metastatic tumors are shown.

Comment_10: Fig 2c: The statistical comparisons appear to be against ES-2 EVs rather than PBS (This also appears to be the case for many of the other figures). Please clarify or correct. The n's for each group need to be written correctly in the legend. The results for c and d appear to show medians not means as stated in Methods. Please clarify.

Response: Thank you for your suggestions; the statistical analysis was indeed unclear for its purpose. The additional statistical analysis (based on PBS as control) was performed as shown in Figure R-a based on your comments. However, in the experiments of Figure 2c, PBS, HOSE1 EV and RMG-1 EV were prepared for control because the purpose of those experiments was to compare the effect of ES-2 EV vs. HOSE1 EV (cancer vs. non-cancer) and ES-2 EV vs. RMG-1 EV (metastatic vs. non-metastatic). To follow these purposes and avoid misunderstandings, further statistical analysis was performed using HOSE1 as a major control, as shown in Figure R-b, which is included in Figure 2c. The description of the n for each group and the bars in graphs were revised according to your comments, and the “statistical analysis” section on page 32 was revised as following texts.

“Unless otherwise stated, the data are presented as the mean \pm s.d, and statistical significance was determined by a Student's t-test. In the graphs of dot plot, the bars indicated median and interquartile range, and statistical significance was determined by a Dunnett's test. $p < 0.05$ was considered statistically significant.”

a

b

Figure. R Additional statistical analysis in the result of Figure 2c

(a) In each graph, the amount of tumor was compared to the group of PBS. Dunnett's test (* $p < 0.05$ and NS = no significance).

(b) In each graph, the amount of tumor was compared to the group of HOSE1 EV. Dunnett's test (* $p < 0.05$ and NS = no significance).

Comment_11: Fig 2d: ** represents a P-value greater than 0.05 and is therefore not significant as stated in Methods. This needs to be changed in the figure and Results section.

Response: As you noted, it was true that there were no differences because we defined $p < 0.05$ as a significant difference in the Methods section. Here, we performed further analysis related to this experiment (Reviewer #3 – Comment_2), and the total number of mice was increased. As a result, the p value was changed to less than 0.05. Figure 2d was substantially revised as according to Figure S, and these data have been included in Figure 2d and Supplementary Figure 3.

Figure. S Reduction in peritoneal metastasis by nSMase2 knock-down

Distribution of photon count in the dissected primary tumors (each left) and the peritoneal metastatic tumors (each right). Orthotopic mouse models were established by injection with A2780 cells and ES-2 cells, and the cells were transfected with the nSMase2 shRNA vector or control vector (TA0726 1C and TA0726 2B). The mice with A2780 cells were euthanized at day 21 and those with ES-2 cells at day 14. KD; knockdown of nSMase2, NC; negative control. Student's t-test (* $p < 0.05$ and NS = no significance).

Comment_12: Fig 3: (c-f) This should include results for EVs from HOSE to determine whether these results are specific to EVs from malignant cells.

Response: We are in total agreement with your suggestion and apologize for the incomplete figures. The results using HOSE1 EV and Mock (PBS) were added to Figure S, and the data were included in Figure 3.

Figure. S Additional images as controls in the *in vivo* experiments

(a-d) The images of HOSE1 EV and PBS were added in Figures 3c-f.

Comment_13: Fig 3d - change "marge" to "merge"

Response: The misspelling in Figure 3d was corrected in response to the Reviewer's comment.

Comment_14: In Fig 4 it is suggested that ES-2 EVs damage the mesothelial MeT-5A cells. The images show a change from cobblestone to spindle shaped cells suggestive of EMT rather than apoptosis or cell damage.

Response: We appreciate your suggestions for improving our data. It is natural to think of EMT from the morphological change in mesothelial cells. As we answered in response to your comment 5, there was no evidence for supporting EMT, and it was obvious that ES-2 EVs induced apoptosis in mesothelial cells from our data. The induction of apoptosis was revealed by the pathway analysis in Figure 5c and caspase assay in Figure 5h.

Comment_15: Fig 5b: Gene expression is compared between mesothelial cells treated with different EVs. However, a mock treated control is not included to understand the basal levels of these genes.

Response: The heat map including the gene expression in mesothelial cells treated with PBS was shown in Figure T, and these data are included in Supplementary Figure 8a. In physiological situations, mesothelial cells were constantly exposed by the EVs from non-cancer cells in ascites. Thus, it is proposed to be preferable to compare cancer EVs to non-cancer EVs.

Figure. T Gene expression analysis in mesothelial cells treated with various EVs and mock.

A heat map showing 89 differentially expressed genes in MeT-5A cells treated with the various EVs and PBS. Three cancer cell lines (ES-2 cell, A2780 cell and SKOV3 cell) vs. HOSE1&2 and Mock. (time point: 48 h, change > 2-fold and $p < 0.05$).

Comment_16: Fig 5h: Caspase activity is compared between EVs from HOSE and ES-2 cells. A mock treated control needs to be included to understand the basal caspase activity of the cells and results presented as absolute values rather than relative values.

Response: We apologize for the lack of a detailed description. The activity of PBS was already subtracted from the activity of HOSE1 and ES-2 in Figure 5h because it was the same as the background signal. The description for this Figure was revised to avoid misunderstandings.

Comment_17: SIFig 2a needs a better explanation.

Response: To avoid confusion, this part was substantially revised as Figure U.

Figure. U Representative bioluminescence images in *in vivo* experiments

Representative images for the results of experiments in Figure 2c. On the left, images of dissected primary tumors are shown, and on the right, those of metastatic tumors are shown.

Comment_18: The title is unclear - perhaps change "toward" to "facilitating"

Response: The title was revised according to the reviewer's comment.

page1, line1

"Malignant extracellular vesicles carrying MMP1 mRNA cause mesothelial cell damage **facilitating** peritoneal dissemination in ovarian cancer"

Comment_19: Minor editing of grammar is suggested.

Response: The entire manuscript was proofread by a native English speaker; the certification is included in the uploaded files.

References

1. Domcke, S., Sinha, R., Levine, D.A., Sander, C. & Schultz, N. Evaluating cell lines as tumour models by comparison of genomic profiles. *Nature communications* **4**, 2126 (2013).
2. Shaw, T.J., Senterman, M.K., Dawson, K., Crane, C.A. & Vanderhyden, B.C. Characterization of intraperitoneal, orthotopic, and metastatic xenograft models of human ovarian cancer. *Mol. Ther.* **10**, 1032-1042 (2004).
3. Slack-Davis, J.K., Atkins, K.A., Harrer, C., Hershey, E.D. & Conaway, M. Vascular cell adhesion molecule-1 is a regulator of ovarian cancer peritoneal metastasis. *Cancer Res.* **69**, 1469-1476 (2009).
4. Sasaki, R., *et al.* Oncogenic transformation of human ovarian surface epithelial cells with defined cellular oncogenes. *Carcinogenesis* **30**, 423-431 (2009).
5. Im, H., *et al.* Label-free detection and molecular profiling of exosomes with a nanoplasmonic sensor. *Nat. Biotechnol.* **32**, 490-495 (2014).
6. Shayesteh, L., *et al.* PIK3CA is implicated as an oncogene in ovarian cancer. *Nat. Genet.* **21**, 99-102 (1999).
7. Zhang, L., *et al.* Genomic and epigenetic alterations deregulate microRNA expression in human epithelial ovarian cancer. *Proc. Natl. Acad. Sci. U. S. A.* **105**, 7004-7009 (2008).
8. Jazaeri, A.A., *et al.* Gene expression profiles of BRCA1-linked, BRCA2-linked, and sporadic ovarian cancers. *J. Natl. Cancer Inst.* **94**, 990-1000 (2002).
9. Witwer, K.W., *et al.* Standardization of sample collection, isolation and analysis methods in extracellular vesicle research. *Journal of extracellular vesicles* **2**(2013).
10. Tominaga, N., *et al.* Brain metastatic cancer cells release microRNA-181c-containing extracellular vesicles capable of destructing blood-brain barrier. *Nature communications* **6**, 6716 (2015).
11. Hoshino, A., *et al.* Tumour exosome integrins determine organotropic metastasis. *Nature* **527**, 329-335 (2015).
12. Costa-Silva, B., *et al.* Pancreatic cancer exosomes initiate pre-metastatic niche formation in the liver. *Nat. Cell Biol.* **17**, 816-826 (2015).
13. Herrera, I., *et al.* Matrix metalloproteinase (MMP)-1 induces lung alveolar epithelial cell migration and proliferation, protects from apoptosis, and represses mitochondrial oxygen consumption. *J. Biol. Chem.* **288**, 25964-25975 (2013).
14. Cauwe, B. & Opdenakker, G. Intracellular substrate cleavage: a novel dimension in the biochemistry, biology and pathology of matrix metalloproteinases. *Crit. Rev. Biochem. Mol. Biol.* **45**, 351-423 (2010).

15. Mannello, F., Luchetti, F., Falcieri, E. & Papa, S. Multiple roles of matrix metalloproteinases during apoptosis. *Apoptosis : an international journal on programmed cell death* **10**, 19-24 (2005).
16. Chen, H., Li, D., Saldeen, T. & Mehta, J.L. TGF-beta 1 attenuates myocardial ischemia-reperfusion injury via inhibition of upregulation of MMP-1. *Am. J. Physiol. Heart Circ. Physiol.* **284**, H1612-1617 (2003).
17. Vos, C.M., *et al.* Cytotoxicity by matrix metalloprotease-1 in organotypic spinal cord and dissociated neuronal cultures. *Exp. Neurol.* **163**, 324-330 (2000).
18. Conant, K., *et al.* Matrix metalloproteinase 1 interacts with neuronal integrins and stimulates dephosphorylation of Akt. *J. Biol. Chem.* **279**, 8056-8062 (2004).
19. Sympson, C.J., *et al.* Targeted expression of stromelysin-1 in mammary gland provides evidence for a role of proteinases in branching morphogenesis and the requirement for an intact basement membrane for tissue-specific gene expression. *J. Cell Biol.* **125**, 681-693 (1994).
20. Powell, W.C., Fingleton, B., Wilson, C.L., Boothby, M. & Matrisian, L.M. The metalloproteinase matrilysin proteolytically generates active soluble Fas ligand and potentiates epithelial cell apoptosis. *Curr. Biol.* **9**, 1441-1447 (1999).
21. Mitsiades, N., Yu, W.H., Poulaki, V., Tsokos, M. & Stamenkovic, I. Matrix metalloproteinase-7-mediated cleavage of Fas ligand protects tumor cells from chemotherapeutic drug cytotoxicity. *Cancer Res.* **61**, 577-581 (2001).
22. Tamsma, J.T., Keizer, H.J. & Meinders, A.E. Pathogenesis of malignant ascites: Starling's law of capillary hemodynamics revisited. *Ann. Oncol.* **12**, 1353-1357 (2001).
23. Zebrowski, B.K., *et al.* Markedly elevated levels of vascular endothelial growth factor in malignant ascites. *Ann. Surg. Oncol.* **6**, 373-378 (1999).
24. Senger, D.R., *et al.* Tumor cells secrete a vascular permeability factor that promotes accumulation of ascites fluid. *Science* **219**, 983-985 (1983).

REVIEWERS' COMMENTS:

Reviewer #1 (Remarks to the Author):

The authors have addressed all comments.

Reviewer #2 (Remarks to the Author):

The authors have addressed my concerns.

Reviewer #3 (Remarks to the Author):

This is a generally well written, easy to follow and interesting manuscript presenting novel results. The majority of comments have been addressed. A few points still require clarification.

Reviewer #3, comment 1. Figures 1a and K suggest that the very large volume of cells has been injected into the fat pad (ballooned appearance) in addition to, or rather than the ovarian bursa which has a very small capacity (eg see PMID: 20811322). This is supported by Fig 1b which appears to show a normal sized ovary in the fat pad (defined by the black line) in the presence of peritoneal metastases. While this does not detract from the results, accuracy is critical. To resolve this, please add histological images (H&E) to show the morphology of the resulting primary tumors.

Fig 2d requires a better description in the legend and text. It appears that both the control and knockdown models have knockdown of SMase2. What then is the difference between NC and KD? Fig 5f: The protein results for MMP1 in the cell lysates do not match the mRNA results that were used to identify MMP1 (Fig 5b). This blot shows that MMP1 protein expression does not change in the mesothelial cells while mRNA expression does. This is a major caveat to the study results and needs to be addressed in the Discussion.

Fig 5i, j: Knockdown of MMP1 is only shown in HMPCs although the original mRNA results are from MTA-5a cells and overexpression is only shown in MTA-5a cells. Caspase results from MMP1 knockdown and overexpression should be shown in both cell lines.

Discussion:

Paragraph 2: endometriosis is not a cell type and this needs to be re-written.

2nd last paragraph: reference to results from personal observations need to be removed unless now published.

As stated above, the finding that MMP1 protein levels in the mesothelial cells do not change in the presence of increased mRNA needs to be discussed / explained.

The Responses to the reviewer #3's comments

Reviewer #3:

We appreciate your important comments, which have made this manuscript further improve. All of the points that you raised have been addressed in the revision.

Comment_1: Figures 1a and K suggest that the very large volume of cells has been injected into the fat pad (ballooned appearance) in addition to, or rather than the ovarian bursa which has a very small capacity (eg see PMID: 20811322). This is supported by Fig 1b which appears to show a normal sized ovary in the fat pad (defined by the black line) in the presence of peritoneal metastases. While this does not detract from the results, accuracy is critical. To resolve this, please add histological images (H&E) to show the morphology of the resulting primary tumors.

Response: It is exactly true that the fat pad has a small capacity. However it is enough to inject 50 μ L of PBS without leakage, as shown in our previous response (Figure K). According to the reviewer's suggestion, histological images (H&E) of each primary tumor were added in Figure 1, as shown in Figure V.

Figure V. Histological images of each primary tumor

(c) Histological features of primary tumors in mouse model. The tumors were obtained from the orthotopic mouse model by using ES-2, A2780, SKOV3 and RMG-1 cells. HE staining. Scale bars, 100 μ m.

Comment_2: Fig 2d requires a better description in the legend and text. It appears that both the control and knockdown models have knockdown of SMase2. What then is the difference between NC and KD?

Response: We apologize for the confusing figure, and appreciate your suggestion. According to your comment, the color of dots in the group of KD and the descriptions for these figures were revised.

Figure 2d:

Legends:

(d) Distribution of photon count in the dissected primary tumors (left) and the peritoneal metastatic tumors (right) using nSMase2 knockdown (KD) cells and negative control (NC) cells. The orthotopic mouse model was established by injection with ES-2 cells, and the cells were transfected with the two different sequences of nSMase2 shRNA vector (KD; #1 and #2) or control vector (NC). The mice were euthanized at day 14. KD; knockdown of nSMase2, NC; negative control. Student's t-test (* $p < 0.05$ and NS = no significance).

Supplementary Figure 3:

Legends:

(b) Distribution of photon count in the dissected primary tumors (left) and the peritoneal metastatic tumors (right) using nSMase2 knockdown (KD) cells and negative control (NC) cells. The orthotopic mouse model was established by injection with A2780 cells, and the cells were transfected with the two different sequence of nSMase2 shRNA vector (KD; #1 and #2) or control vector (NC). The mice were euthanized at day 30. KD; knockdown of nSMase2, NC; negative control. Student's t-test ($*p < 0.05$ and NS = no significance).

Comment_3: Fig 5f: The protein results for MMP1 in the cell lysates do not match the mRNA results that were used to identify MMP1 (Fig 5b). This blot shows that MMP1 protein expression does not change in the mesothelial cells while mRNA expression does. This is a major caveat to the study results and needs to be addressed in the Discussion.

Response: As you commented, the difference of MMP1 protein level in mesothelial cell treated with EVs was not the same as the results of mRNA level. However MMP1 is a type of secretory protein, and it is functional at extracellular matrix. Therefore the amount of MMP1 protein level in cell culture supernatants is often analyzed (PMID: 15927440, PMID: 19608765). As the result of western blotting analysis which is knockdown experiments for MMP1 (Supplementary Figure 10d), MMP1 protein level in cell lysates was not so changed, but that in culture supernatant was significantly decreased. In other words, the difference might tend to appear in culture supernatant. The increase of MMP1 mRNA in mesothelial cells caused by direct transfer via EVs, and the part of the mRNA could be translated. For these reasons, it is supposed that the difference of the results between protein and mRNA levels in cells was observed. The following sentences were added in the Discussion section.

“The mRNA was translated in mesothelial cells (Figure 5f). However the difference of MMP1 protein level in cell lysates was not observed, because MMP1 is a type of secretory protein which is often analyzed in cell culture supernatants^{42,43}. In addition, the difference of protein level was not the same as the results of mRNA (Supplementary Figure 9), presumably because the increase of MMP1 mRNA in mesothelial cells was caused by the direct transfer via EVs, and the part of the mRNA was translated.”

Comment_4: Fig 5i, j: Knockdown of MMP1 is only shown in HMPCs although the original mRNA results are from MTA-5a cells and overexpression is only shown in MTA-5a cells. Caspase results from MMP1 knockdown and overexpression should be shown in both cell lines.

Response: Further experiments about Knockdown of MMP1 were performed under the same condition by using MeT-5A cells, and the result was shown in Figure W. This result was included in Figure 5i. On the other hand, it was impossible to perform further experiments about overexpression of MMP1 under the same condition by using HPMCs because the cell is the type of primary culture cell.

Figure W. Caspase activity in Met-5A cells treated with EVs derived from the MMP1 shRNA-expressing ES-2 cells (knockdown: K.D.) or control (negative control: NC). Error bars represent s.d., Student's t-test (* $p < 0.01$).

Comment_5: Discussion: Paragraph 2: endometriosis is not a cell type and this needs to be re-written.

Response: We totally agree to this comment. To avoid misunderstandings, the parts were revised according to the comments as the following sentences.

“For these subtypes, the cells of origin for the ovarian cancer cells are derived from non-ovarian tissues, i.e., the origin of the high-grade serous type is a distal fallopian tube, and that of **the clear-cell type is an endometrial cyst**, which is a common benign gynecological tumor.”

Comment_6: 2nd last paragraph: reference to results from personal observations need to be removed unless now published.

Response: This work is now in press (Nishida-Aoki, N., et al. Disruption of circulating extracellular vesicles as a novel therapeutic strategy against cancer metastasis. *Mol. Ther.* (2017), DOI information: 10.1016/j.ymthe.2016.10.009.). The part was partially revised as the following sentence. We would like to ask the editor to be able to wait a little more.

“As **our recent report**, we disrupted cancer-derived EVs by therapeutic antibody administration in a metastatic cancer mouse model and drastically suppressed metastatic cells ⁶².”

Comment_7: As stated above, the finding that MMP1 protein levels in the mesothelial cells do not changes in the presence of increased mRNA needs to be discussed / explained.

Response: This comment have been addressed in the response in the **Comment_3**.